# Neural pathways and computations that achieve stable contrast processing tuned to natural scenes

Burak Gür [1,2], Luisa Ramirez[1], Jacqueline Cornean[1], Freya Thurn [1], Sebastian Molina-Obando [1], Giordano Ramos-Traslosheros[1,3] & Marion Silies [1] ✉

Natural scenes are highly dynamic, challenging the reliability of visual processing. Yet, humans and many animals perform accurate visual behaviors, whereas computer vision devices struggle with rapidly changing background luminance. How does animal vision achieve this? Here, we reveal the algorithms and mechanisms of rapid luminance gain control in *Drosophila*, resulting in stable visual processing. We identify specific transmedullary neurons as the site of luminance gain control, which pass this property to direction-selective cells. The circuitry further involves wide-field neurons, matching computational predictions that local spatial pooling drive optimal contrast processing in natural scenes when light conditions change rapidly. Experiments and theory argue that a spatially pooled luminance signal achieves luminance gain control via divisive normalization. This process relies on shunting inhibition using the glutamate-gated chloride channel GluClα. Our work describes how the fly robustly processes visual information in dynamically changing natural scenes, a common challenge of all visual systems.

Many animals heavily rely on visual cues to navigate their environment. To achieve stable visual processing, visual systems must work reliably in many different contexts, for example when light conditions change. We encounter rapid changes in the visual input when moving through the environment, or simply when our gaze follows an object from a sunny spot into a shaded area[1–3]. Even beyond living systems, rapid illumination changes challenge the information processing of computer-vision algorithms, such as those implemented in camera-based navigation systems[4,5]. Hence, many self-driving car systems rely on additional radar or lidar-based navigation to properly separate the contrast of an object from its background. However, animal eyes can do without such technology. What can we learn from animals to stably process visual cues under constantly changing lighting conditions? Here, we investigate the neuronal principles of

information processing under dynamic luminance conditions typical in natural environments.

Rapid changes in luminance within a neuron's receptive field (RF) can occur within hundreds of milliseconds, for example due to eye, head, or body movements, but also due to movement of an object. All these challenge the accuracy of visual processing (Fig. 1a), because contrast is computed as the change in luminance, relative to background luminance, which changes when light conditions change. Yet, visual systems of animals from humans to fruit flies maintain stable contrast perception, arguing that a fast post-receptor luminance gain control mechanism acts in addition to slower gain control mechanisms starting in photoreceptors[2,6–8]. Evidence for such a mechanism exists along the visual hierarchy and in multiple species. For instance, cat Y-type retinal ganglion cells (RGC) exhibit rapid luminance gain con-

[1]Institute of Developmental Biology and Neurobiology, Johannes-Gutenberg University Mainz, Mainz, Germany. [2]Present address: The Friedrich Miescher Institute for Biomedical Research (FMI), Basel, Switzerland. [3]Present address: Department of Neurobiology, Harvard Medical School, Boston, MA, USA. ✉e-mail: msilies@uni-mainz.de

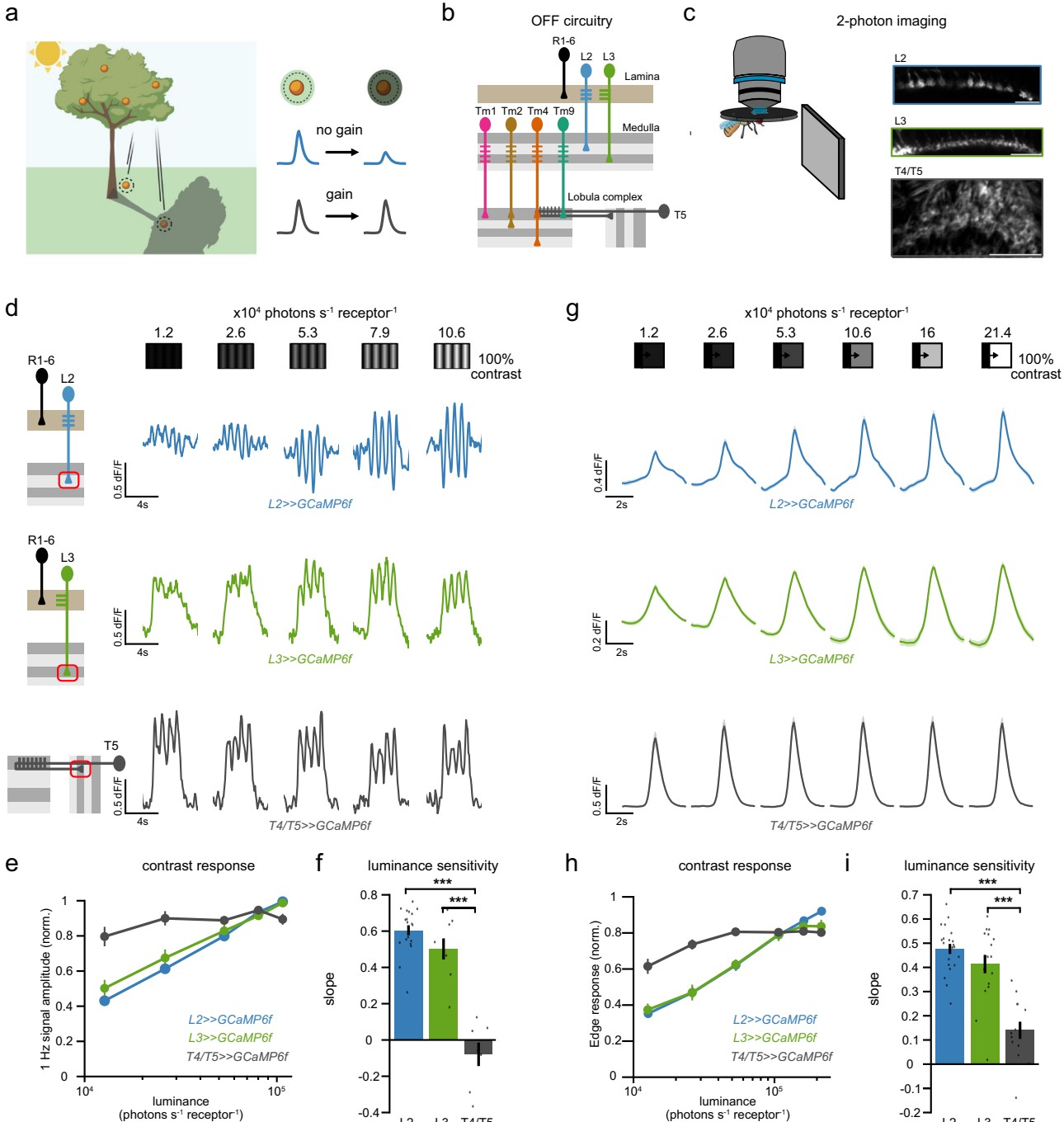

**Fig. 1 | Post-receptor luminance gain control ensures stable motion responses irrespective of fast changing luminances. a** Rapid luminance gain control ensures stable contrast representation when luminance changes rapidly. Upon a fast change of a neuron's receptive field (RF) to a differently lit region, a neuron without luminance gain control will represent the contrast between an object (orange) and its background differently, depending on background luminance (blue traces). Fast luminance gain control adjusts the neuron's contrast responses according to background luminance, keeping contrast representations stable across luminance (black traces). Created with biorender.com. **b** Schematic of the fly visual OFF circuits. Photoreceptors synapse onto lamina neurons L2 and L3 which give inputs to the medulla neurons Tm1, Tm2, Tm4, and Tm9 that in turn converge onto the dendrites of the direction-selective cell T5. **c** In vivo two photon images of axon terminals of L2, L3 and T4/T5 neurons expressing the calcium indicator GCaMP6f. Scale bars are 17μm. **d-f** Calcium imaging of L2 (blue), L3 (green) and T4/T5 neurons (gray) while stimulating the fly eye with drifting 1 Hz gratings of constant 100% Michelson contrast and five rapidly changing luminances. **d** Representative calcium

responses of a single axon terminal to the stimulus for each neuron type. **e** Mean normalized contrast responses (F1 amplitude) of each neuron across luminances. **f** Slopes of the contrast responses, obtained from a linear fit to the data shown in (**e**), depicting the log-luminance dependence of each neuron. ***$p < 0.001$, one way ANOVA with post-hoc Tukey HSD test. $P$ values for pairwise comparisons: L2–L3 = 0.2508, L2- T4/T5 = 4.8852e-11, L3- T4/T5 = 6.1906e-08. **g-i** Calcium imaging of L2 (blue), L3 (green) and T5 neurons (gray) while stimulating the fly eye with moving OFF edges of −100% Weber contrast varying in background luminance. **g** Mean calcium responses of each neuron type after aligning the response peaks of each axon terminal. **h** Contrast responses (peak response) of each neuron across luminances. **i** Slopes of the contrast responses obtained from a linear fit to the data shown in (**g**). ***$p < 0.001$, one way ANOVA with post-hoc Tukey HSD test. $P$ values for pairwise comparisons: L2–L3 = 0.3154, L2–T5 = 5.4996e-09, L3–T5 = 2.1441e-06. Error bars and patches represent ±SEM. Means are calculated across flies. Sample sizes for (**d-f**), L2: $n = 19(141)$, L3: $n = 8(150)$, T4/T5: $n = 7(362)$; (**g-i**), L2: $n = 21(158)$, L3: $n = 16(238)$, T5: $n = 13(765)$. Sample sizes are given as: flies(cells).

trol under dim light conditions[8] and lateral geniculate nucleus (LGN) neurons stably compute contrast irrespective of fast luminance variations[2]. Similarly, humans can accurately estimate contrast within hundreds of milliseconds, when luminance conditions change[7,9], a property which we refer to as luminance invariance. Similarly, visually guided behaviors in flies are luminance-invariant at rapid time scales[6,10,11]. However, the full neural implementation and the cellular and circuit mechanisms underlying this post-receptor luminance gain control are not known in any visual system.

A common way to keep representations of an environmental feature stable, or independent from other features is normalization[12–14]. In this case of gain control neural responses are scaled by the pooled activity of neighboring neurons[15]. This canonical computation is found in various forms in sensory systems[12], but also in computer vision algorithms, such as convolutional neural networks, where normalization improves their generalization[16,17]. Although it is not known if and how normalization is implemented to keep contrast representations in rapidly changing environments stable, recent studies in *Drosophila melanogaster* provided first insights into the neuronal mechanisms underlying rapid luminance gain control. Flies show stable behavioral responses to both ON and OFF contrasts in rapidly changing luminance environments[6,10]. This depends on luminance-sensitive signals downstream of photoreceptors, especially from L3 lamina monopolar cells (LMCs) (Fig. 1b)[6,11,18]. Different LMCs (L1, L2, L3) provide input to medulla neurons, where further processing of the visual information in distinct ON and OFF pathways enables the extraction of direction-selective (DS) signals in the T4 (ON-DS) and T5 (OFF-DS) neurons[19–22]. T4/T5 neurons are required for different types of motion-evoked behaviors including optomotor responses[23–25], escape behaviors elicited by looming stimuli[26,27], and course stabilization[28,29]. Whereas this circuitry is computing information locally, for a gain control operation that adjusts for background luminance changes, it is feasible that more global computations are in place. Conversely, given that luminance correlations in natural scenes rapidly drop with distance[1], too global could be detrimental, potentially explaining why cell phone cameras which adjust gain of the whole scene using a single value often fail at capturing structure (contrast) across the scene. Neurons that could capture information across a specific, wider region of visual space, exist in vertebrate and invertebrate visual systems, such as different types of horizontally projecting wide-field neurons[30–32]. However, whether more global computations play a role in luminance gain control, whether there is a 'sweet spot' of the spatial extent of luminance gain control, and how information from columnar and wide-field neurons is combined to achieve stable visual processing is entirely unexplored.

Here we investigate the computational principles, as well as the circuit and molecular mechanisms underlying rapid luminance gain control in the fly visual system. Using in vivo two-photon imaging, we identify the dendrites of two third-order neurons, Tm1 and Tm9, as the site at which luminance gain control is first implemented. A computational analysis of a set of natural scenes under different luminance conditions, and as viewed by a moving animal, then demonstrates the need for local spatial pooling of background luminance. Motivated by this finding, and by combining state-of-the-art connectomics and functional imaging, we show that luminance information is spatially pooled by Tm neurons and identify Dm12 as the wide-field input. Finally, a biophysical model suggests that shunting inhibition from wide-field neurons drives the normalization step of rapid luminance gain control, for which genetic manipulations revealed the inhibitory glutamate-gated chloride channel GluClα as the relevant substrate. Together, our data reveal the computational, algorithmic, and mechanistic implementation of rapid luminance gain control, unraveling how visual information can robustly be processed in dynamically changing natural visual scenes.

## Results

### Post-receptor luminance gain control ensures stable motion responses irrespective of fast-changing luminances

To identify where in the visual circuitry luminance-invariant responses are first established, we used in vivo two-photon imaging to record neuronal responses along the hierarchy of the OFF pathway (Fig. 1b). We first imaged the axon terminals of L2 or L3 neurons expressing the genetically encoded calcium indicator GCaMP6f[33] while presenting visual stimuli to the fly (Fig. 1c). To measure contrast computation in changing luminance environments, we showed the flies moving sinusoidal gratings of a constant contrast (100% Michelson) at five rapidly changing luminances (Fig. 1d). Both L2 and L3 responded with oscillating calcium signals matching the temporal frequency of the sinusoidal grating (Fig. 1d). The F1 amplitude of these responses, which we refer to as the contrast response, changed with mean luminance in both L2 and L3, demonstrating that LMCs cannot stably estimate contrast in rapidly changing luminances.

Since motion information should be extracted independently of environmental luminance fluctuations, we next asked if the direction-selective T4/T5 neurons compute contrast stably. T4/T5 axon terminals responded similarly to the same contrast in rapidly changing luminances (Fig. 1d, e). We quantified the luminance dependency of each neuron type by computing the slopes of the contrast response curves (Fig. 1f). T4/T5 depended significantly less on luminance than the LMCs, demonstrating the presence of luminance gain control (Fig. 1e, f).

We next used OFF edges moving onto backgrounds of varying luminances to stimulate LMCs and the OFF-DS T5 cells (Fig. 1g–i). Again, the responses of T5 neurons but not of the LMCs were luminance-invariant (Fig. 1h, i). Thus, the luminance gain control measured in DS neurons persists across different stimulus conditions. A rapid luminance gain control is implemented in visual circuitry between the LMCs and DS neurons, enabling stable contrast responses in rapidly changing illuminations.

### Luminance gain control arises pre-synaptically to direction-selective cells in distinct medulla neurons

The third-order medulla neurons Tm1, Tm2, Tm4, and Tm9 connect LMCs to the dendrites of the T5 cells[34–36] (Fig. 1b) making them candidates for the emergence of luminance gain control. We measured calcium signals in their axon terminals while showing the fly sinusoidal gratings of 100% contrast and changing luminance. Similar to LMCs, Tm2 and Tm4 contrast responses varied strongly with luminance (ANOVA $p < 0.05$, Fig. 2a–c). Tm1 and Tm9 neurons instead exhibited highly similar contrast responses in different luminances, (Fig. 2d), and they were significantly less luminance dependent than their corresponding LMC inputs, suggesting that rapid luminance gain control emerges within Tm1 and Tm9 (Fig. 2e, f). Qualitatively, the gain in these two cell types differed: Tm1 neurons were luminance invariant, whereas Tm9 neurons even enhanced responses to low luminances (negative slope, Fig. 2f), suggesting that Tm1 and Tm9 employ distinct mechanisms to implement luminance gain control.

Animals are exposed to a broad range of contrasts while navigating natural scenes[3]. We thus imaged the responses of LMCs, Tm1, and Tm9 to different contrasts ranging from 20% to 100%, in varying luminance conditions (Fig. 2g). LMC responses to all contrasts increased with mean luminance (Fig. 2h and Supplementary Fig. 1), as also reflected in the negative slopes of the iso-response contour lines in the contrast-luminance maps of L2 indicating the absence of luminance gain control in these cells (Fig. 2h). Tm1 and Tm9 neurons instead exhibited gain-corrected responses for all contrasts measured (Fig. 2i–l): Tm1 axon terminals were luminance-invariant across all contrasts (Fig. 2i, ANOVA $p > 0.05$). Tm9 neuron axon terminals were luminance invariant for 80% and 100% contrast

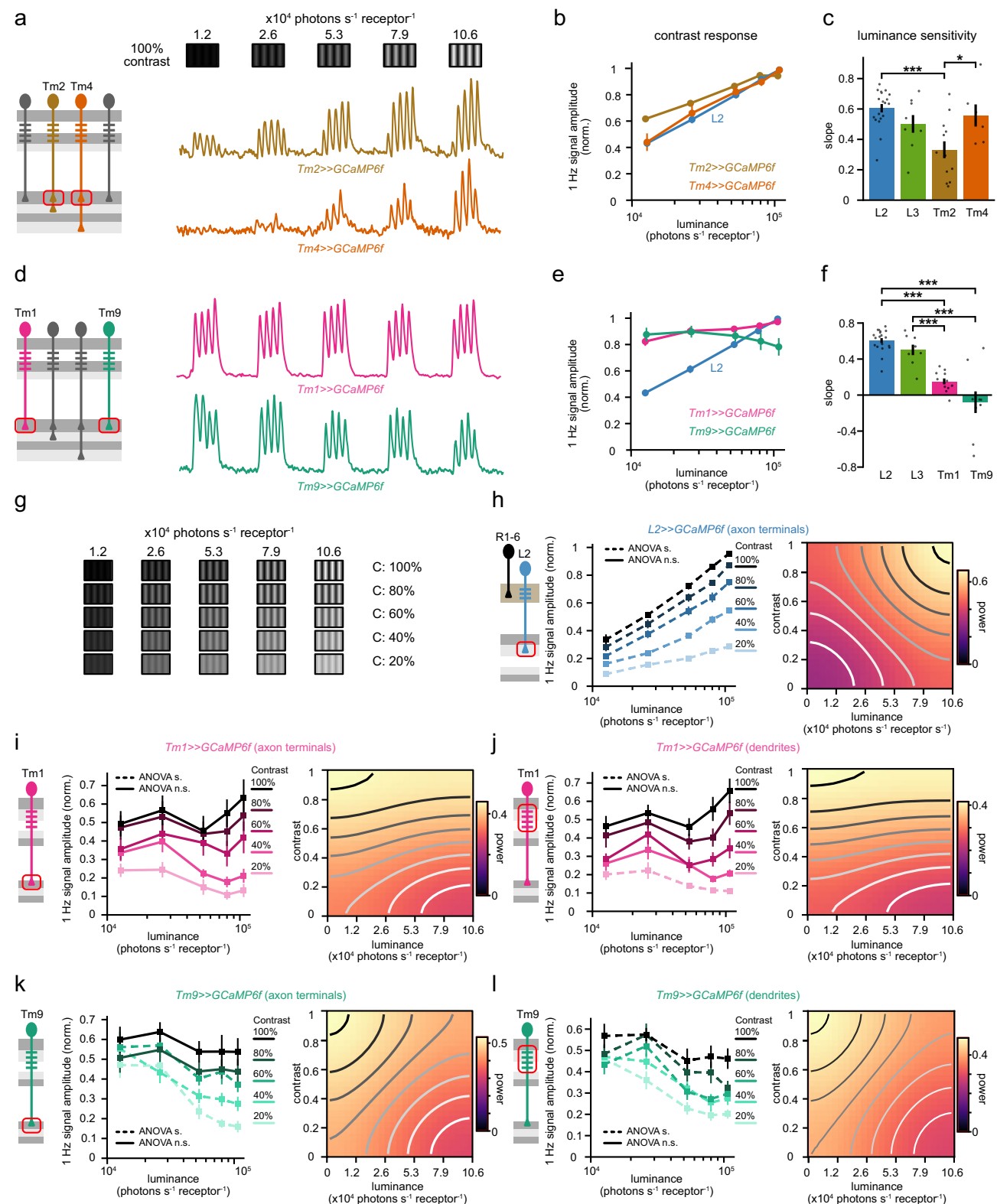

(Fig. 2k, ANOVA $p > 0.05$), and at low contrast the dependency on luminance was inverted with respect to L2, with higher response at low luminances (Fig. 2k, ANOVA $p < 0.05$). The luminance gain control in dendrites was similar to the one measured in axon terminals. Tm1 dendritic calcium signals were luminance-invariant for all contrasts except 20% Michelson contrast, and Tm9 dendritic signals enhanced responses at lower luminance to all contrasts (Fig. 2j, l). Taken together, rapid luminance gain control first arises in the dendrites of Tm1 and Tm9, establishing distinct contrast responses of both neurons.

## Local pooling of luminance leads to robust estimates of contrast in natural scenes

Computing contrast from dynamic visual scenes, e.g., while navigating, requires normalizing point luminance values by background luminance. Because luminance correlations across natural scenes quickly

**Fig. 2 | Luminance gain control arises pre-synaptically to DS cells in distinct medulla neurons. a** Representative calcium responses of a single axon terminal to drifting 1 Hz gratings of constant 100% Michelson contrast and changing luminances for Tm2 (brown) and Tm4 (orange). **b** Mean normalized contrast responses (F1 amplitude) of each neuron across luminances. **c** Slopes of the contrast responses depicting the log-luminance dependence for each neuron. *$p < 0.05$, ***$p < 0.001$, one way ANOVA with post-hoc Tukey HSD test. *P* values for pairwise comparisons: L2-Tm2 = 0.0003, L2-Tm4 = 0.9284, L3-Tm2 = 0.1280, L2-Tm2 = 0.9251, Tm2-Tm4 = 0.0453. **d–f** Same as (**a–c**) for Tm1 (magenta) and Tm9 (teal) neurons. *P* values for pairwise comparisons: L2-Tm1 = 3.4936e-06, L2-Tm9 = 1.1560e-09, L3-Tm1 = 0.0034, L2-Tm9 = 3.4945e-06, Tm1-Tm9 = 0.0716. **g–l** Calcium imaging of L2 (blue), Tm1 (magenta) and Tm9 (teal) neurons while stimulating the fly with drifting 1 Hz gratings of changing contrast and luminance. **g** The stimulus set consisted of 5 different contrasts and 5 different luminances. **h** Left: Mean contrast responses of L2 neurons across luminance, each curve represents responses to one grating contrast. One way ANOVA between the luminances for each contrast to assess luminance-invariant responses. Dashed lines represent $p < 0.05$ and solid lines represent $p > 0.05$. P values for ANOVA: L2-contrast:0.2 = 1.5524e-11, L2-contrast:0.4 = 1.7852e-14, L2-contrast:0.6 = 5.5199e-16, L2-contrast:0.8 = 6.6147e-17, L2-contrast:1 = 1.9576e-16. Right: Heatmap of contrast responses as a function of luminance and contrast. Isoresponse lines visualize the dependency of responses to contrast and luminance. **i** Same as (**h**) for Tm1 axon terminals. P values for ANOVA: contrast:0.2 = 0.2580, contrast:0.4 = 0.0525, contrast:0.6 = 0.6693, contrast:0.8 = 0.8489, contrast:1 = 0.6517. **j** Tm1 dendrites. P values for ANOVA: contrast:0.2 = 0.0448, contrast:0.4 = 0.0914, contrast:0.6 = 0.0953, contrast:0.8 = 0.3013, contrast:1 = 0.2256. **k** Tm9 axon terminals. P values for ANOVA: contrast:0.2 = 9.3020e-05, contrast:0.4 = 0.0019, contrast:0.6 = 0.0249, contrast:0.8 = 0.3048, contrast:1 = 0.4134. **l** Tm9 dendrites. *P* values for ANOVA: contrast:0.2 = 2.7711e-06, contrast:0.4 = 8.0993e-05, contrast:0.6 = 3.1758e-06, contrast:0.8 = 0.0266, contrast:1 = 0.0419. Error bars represent ±SEM. Means are calculated across flies. Sample sizes are given as # flies(cells) and are (**a–f**), L2: *n* = 19(141), Tm2: *n* = 12(186), Tm4: *n* = 6(59), Tm1: *n* = 11(148), *n* = Tm9: 9(111). (**g–l**) L2 axon terminals: *n* = 7(103), Tm1 dendrites: *n* = 8(102), Tm1 axon terminals: *n* = 8(82), Tm9 dendrites: *n* = 7(83), Tm9 axon terminals: *n* = 7(92).

drop with distance[1,37], estimating a reliable background should require a local computation, e.g., pooling across near spatial locations. To quantify the performance of contrast estimation when implementing luminance gain control via local spatial pooling, we modeled neuronal responses to a natural scene under differently lit conditions[37] (Fig. 3a, Supplementary Fig. 2c). We implemented a voltage-membrane model with its parameters obtained by fitting LMC contrast responses to moving gratings (Supplementary Fig. 2a, b and Methods). This model predicted how LMCs (e.g., L2) would encode these natural scenes (Fig. 3b). We simulated random flight trajectories to sample neuronal contrast responses under dynamic conditions (Fig. 3b, c). LMC contrast responses differed between shaded and sunny conditions: the sunny condition showed a bimodal contrast distribution that was shifted to brighter luminances, owed to the strong luminance dependence of LMCs (Fig. 3c, Supplementary Fig. 2d). Thus, changes in luminance challenge stable contrast estimation in natural scenes.

The goal of the visual system should be to keep contrast representations stable, i.e., to implement a mechanism that minimizes differences in these contrast distributions across luminance conditions. One possible mechanism is to use a spatially pooled luminance signal as a normalization factor that scales neuronal responses proportionally to the local luminance information. To quantify contrast responses with such a luminance gain control, we normalized the signal downstream of LMCs by a luminance background computed from neighboring spatial locations (Fig. 3d, e). When the spatial pooling was minimal (few pixels), the contrast distributions from the different luminance conditions converged but lost structure and became very narrow, leading to poor contrast estimation (Fig. 3d, e). Increasing the pooling size preserved the contrast resolution of the image in addition to keeping it consistent across different luminance conditions (Fig. 3d, e). Together, our findings suggest the existence of a trade-off between contrast information structure and stable contrast estimation under different luminance conditions which is determined by the spatial pooling size. To assess this size dependence quantitatively, we defined a loss function that minimizes differences in contrast responses across luminance conditions while maximizing contrast information content (Methods). For the natural scene analyzed, a wide range of local pooling sizes conveyed reliable contrast computation (Fig. 3f, Supplementary Fig. 2e). However, the large variance of the cost function at narrow pooling sizes suggested a larger sensitivity to the type of luminance change in the images. The same analysis done on another natural scene under differently lit conditions, but at a coarser spatial scale (Supplementary Fig. 2f, g) revealed scene-specific differences at very narrow spatial pooling extents, where the loss function was higher (Fig. 3g, h). Here, large pooling sizes had a more negative effect than in the previous image, probably because of a stronger decorrelation of luminance in this image. However, in both scenes,

local spatial pooling optimizes both contrast resolution and reliability (Fig. 3h). This highlights that local spatial pooling establishes reliable contrast estimation across the visual field in natural scenes. Although the optimal pooling sizes might vary across scenes, the existence of such a trade-off remains valid.

## Spatial pooling is necessary for luminance gain control

Motivated by the natural scene analysis, we next tested if the first neurons to implement rapid luminance gain control, Tm1 and Tm9, use spatial pooling. To probe for spatial pooling in the context of luminance gain control, we developed a stimulus paradigm in which we first mapped the neurons' RF centers using white noise stimuli. We then centered a moving sinusoidal grating on a single neuron's RF (Fig. 4a). We first used 5° wide moving gratings of constant contrast and luminance, designed to stimulate a single neuron, and added annuli of varying sizes and luminances (Fig. 4b). Only if the neuron implemented a spatial pooling mechanism to achieve luminance gain control, the contrast responses to the grating would change with annulus size and luminance. The contrast response of single Tm1 or Tm9 neurons increased when the annulus luminance was lower than the intermediate background and decreased when the annulus luminance was higher. The responses also depended on annulus size. Increasing the size of the annulus led to changing contrast responses, with maximum modulation occurring around 10–15° of visual space (Fig. 4c, d, Supplementary Fig. 3). This suggests that luminance gain control is implemented via narrow spatial pooling as also predicted by the natural scene-based model (Fig. 3).

We next used gratings of different sizes (5–30°) and background luminances to study how this affects luminance gain control properties (Fig. 4e). Both Tm1 and Tm9 neurons responded to the contrast of those gratings, and their luminance dependency varied with size. Tm1 contrast responses increased with luminance when the gratings were small (Fig. 4f) and became luminance-invariant at a diameter of 20–25° (ANOVA $p > 0.05$, Fig. 4h). Tm9 contrast responses were luminance-invariant at small grating sizes (Fig. 4g, i, ANOVA $p > 0.05$), and the slope of the contrast responses decreased from smaller to larger gratings, and eventually became negative, showing that Tm9 enhances responses at low luminances (Fig. 4g, i, ANOVA $p < 0.05$). In summary, both Tm1 and Tm9 require local spatial pooling to achieve luminance gain control with some stimulus-specific effects on the pooling extent.

## A normalization model using shunting inhibition explains luminance gain control in medulla neurons

Neuronal responses are scaled by the local background luminance to achieve stable contrast representations. To investigate the biological implementation, we implemented a circuit model. To model Tm1 responses, we included a wide-field neuron into the circuit that pools

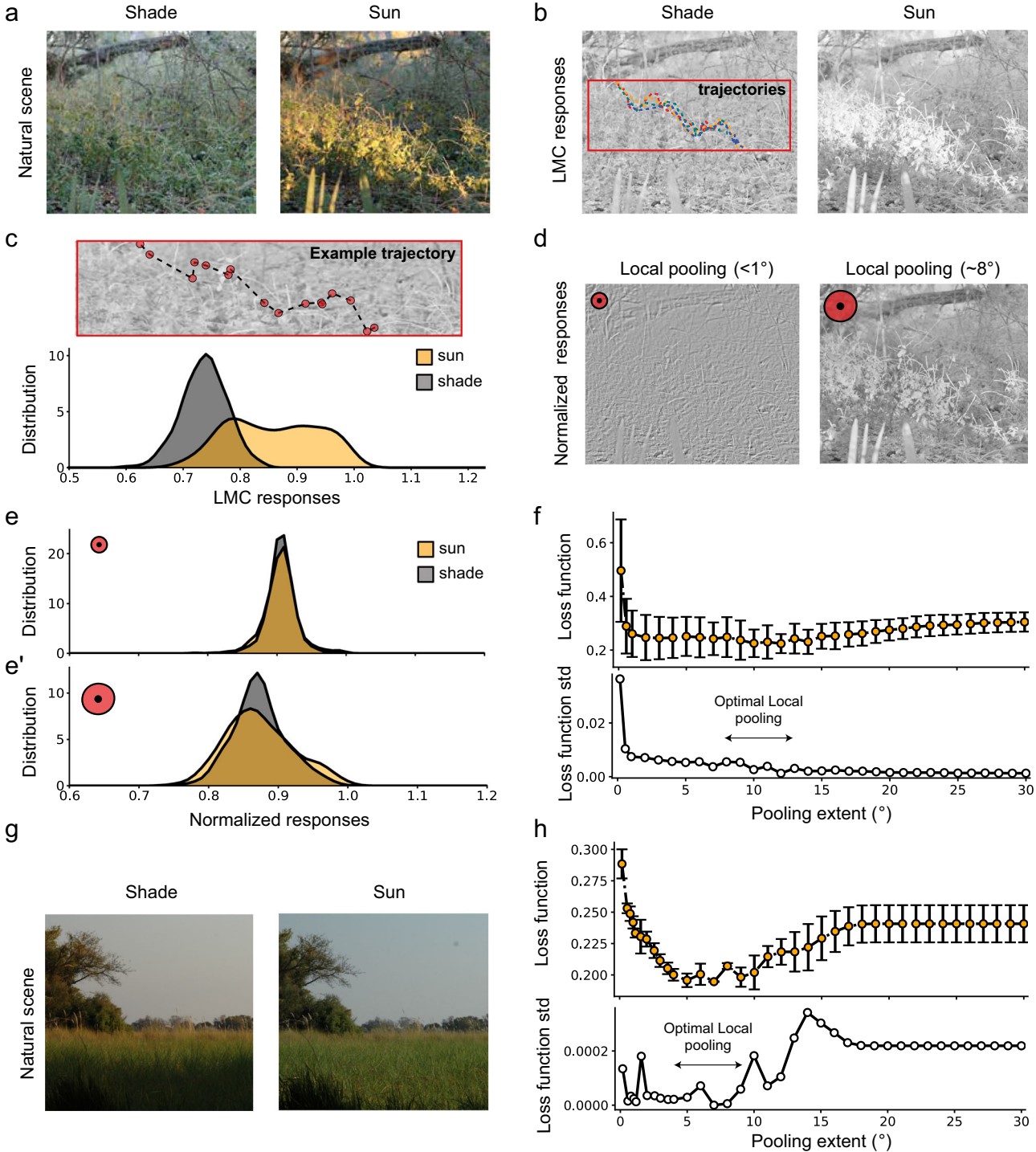

**Fig. 3 | Local pooling of luminance can lead to robust estimates of contrast in natural scenes. a** Example of a natural scene from[37] in shaded and sunny conditions used to model contrast responses of LMC neurons (Methods). **b** LMC responses under two luminance conditions simulated using a voltage-membrane-potential model. Each pixel of the scene is considered a luminance input to the neuron (Methods). Inset: Examples of simulated trajectories on a section of the visual scene. **c** Top: Example of a fly trajectory used to sample LMC contrast responses in the two images. Below: Probability distribution of LMC contrast responses in both shaded and sunny conditions for 15 simulated stochastic trajectories (Methods). **d** Normalized responses downstream of LMCs under sunny conditions. Left: Normalization with a local pooling smaller than 1° of visual angle. Right: Normalization with a local pooling of 8°. **e–e'** Probability distribution of

normalized responses. **e** Pooling size smaller than 1°. The distributions from both luminance conditions have a large overlap but a significant reduction in the response range compared to the non-normalized distributions. **e'** Pooling size of 8°. The distributions in both luminance conditions overlap and preserve the range of responses. **f** Top: Loss function (Methods) as a function of pooling extent, calculated from 4 different luminance conditions of the same natural scene using the distribution of 15 trajectories ($n = 4$ luminance conditions)[37] (Supplementary Fig. 2). Orange markers and error bars show mean ± standard deviation (std). Bottom: Std of the loss function as a function of pooling extent. **g** Example of a different natural scene under two luminance conditions. **h** Same as (**f**) for the natural scene shown in (**g**) ($n = 2$ luminance conditions).

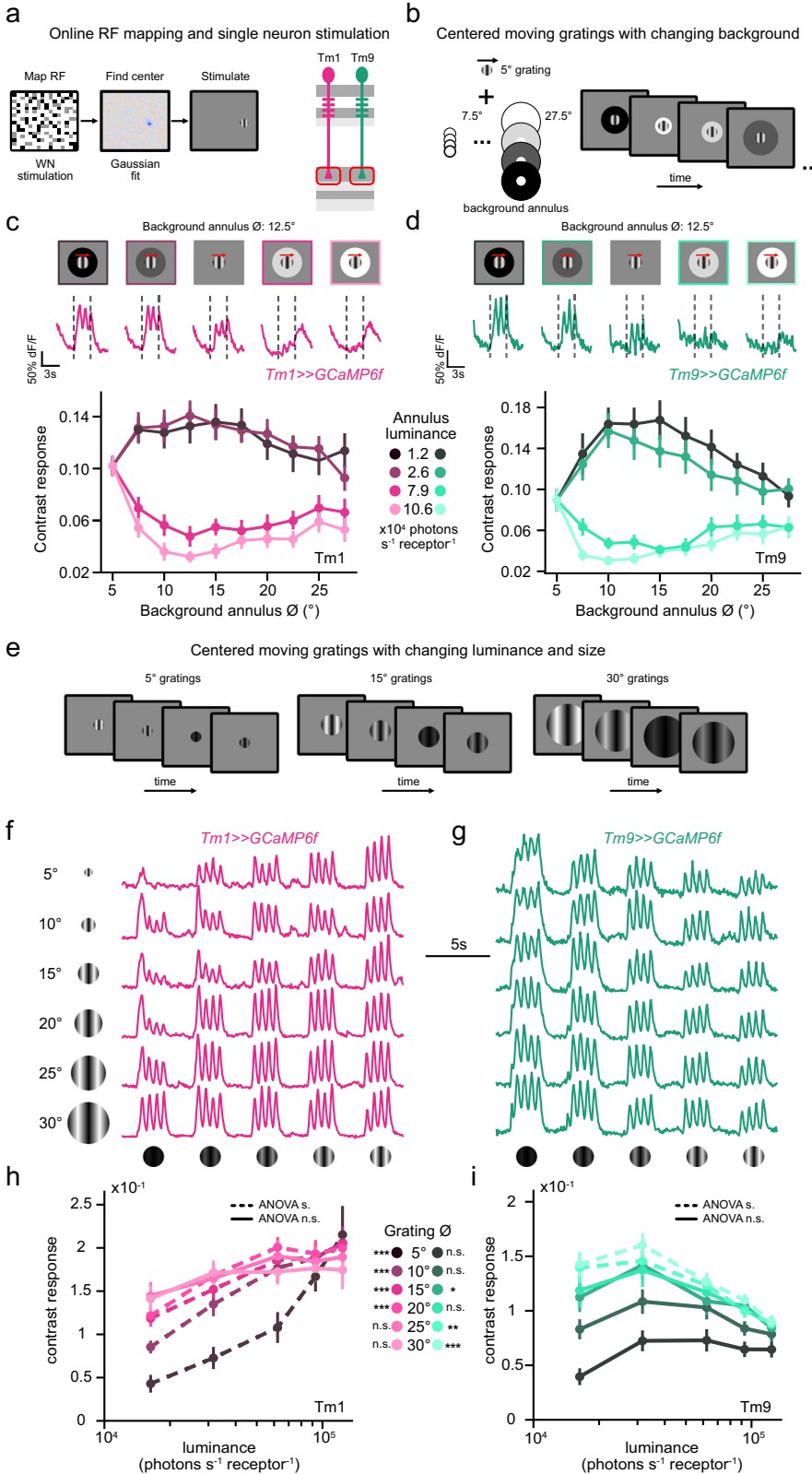

responses from Tm1's major presynaptic input L2 neurons from multiple columns within a region of a given diameter (Fig. 5a). We then computed the response of Tm1 as being proportional to the input from L2 and divided by the signal of the hypothetical wide-field neuron using shunting inhibition as biophysical implementation (Fig. 5a)[38,39]. The model fitted the experimentally observed neuronal responses, using a pooling size of 15°, suggesting that normalization can be the

computation underlying luminance gain control in Tm1 (Fig. 5b, c and "Methods"). Next, we modeled Tm9 contrast responses by implementing a similar circuit model, only that the main input to Tm9 was provided by L3 neurons (Fig. 5d). Interestingly, yielding the sign shift of the luminance-dependency specific to Tm9 required a non-linear normalization (Fig. 5d–f and "Methods"). Specifically, we considered the normalization factor from the wide-field neuron to be a non-linear

**Fig. 4 | Spatial pooling is necessary for luminance gain control. a** Receptive fields (RF) of individual axon terminals were mapped via ternary white noise consisting of 2.5° squares. **b** The RF center of a single axon terminal was stimulated with a circular 5° sinusoidal grating moving at 1 Hz with constant contrast (100% Michelson) and constant luminance. The grating contained a background annulus of changing diameters (7.5–27.5°) and changing luminances. **c** Top: Example calcium responses of a single Tm1 axon terminal (magenta) to the stimulus with a background annulus of 12.5°. Dashed lines represent the time of stimulus presentation. Bottom: Mean contrast responses (F1 amplitude) across background annulus diameters. Each curve represents a different background annulus luminance. **d** Same as (**c**) for Tm9 neurons (teal). **e** Drifting gratings centered at the RF center with constant contrast and changing luminances at varying diameters (5–30°). **f, g** Calcium responses of a single Tm1 axon terminal (F, magenta) and Tm9

axon terminal (g, teal). **h, i** Mean contrast responses (F1 amplitude) across luminance for Tm1 (**h**, magenta) and Tm9 (**i**, teal) neurons. Each curve represents a different grating diameter. One way ANOVA between the luminances for each contrast to assess luminance-invariant responses. Dashed lines represent $p < 0.05$ and solid lines represent $p > 0.05$. ANOVA results are also provided at the grating diameter legend panel as n.s.: not significant, *$p < 0.05$, **$p < 0.01$, ***$p < 0.001$. $P$ values for ANOVA for Tm1 neurons: size: 5° = 1.14278e-06, size: 10° = 9.69355E-06, size: 15° = 1.75234E-05, size: 20° = 0.0007, size: 25° = 0.3124, size: 30° = 0.5898. $P$ values for ANOVA for Tm9 neurons: size: 5° = 0.0973, size: 10° = 0.1156, size: 15° = 0.0288, size: 20° = 0.1049, size: 25° = 0.0028, size: 30° = 2.00E-05. Error bars represent ±SEM. Means are calculated across cells. Sample sizes in (**c, d**), Tm1: $n = 14$, Tm9: $n = 15$ cells. (**f–i**) Tm1: $n = 12$, Tm9: $n = 11$ cells.

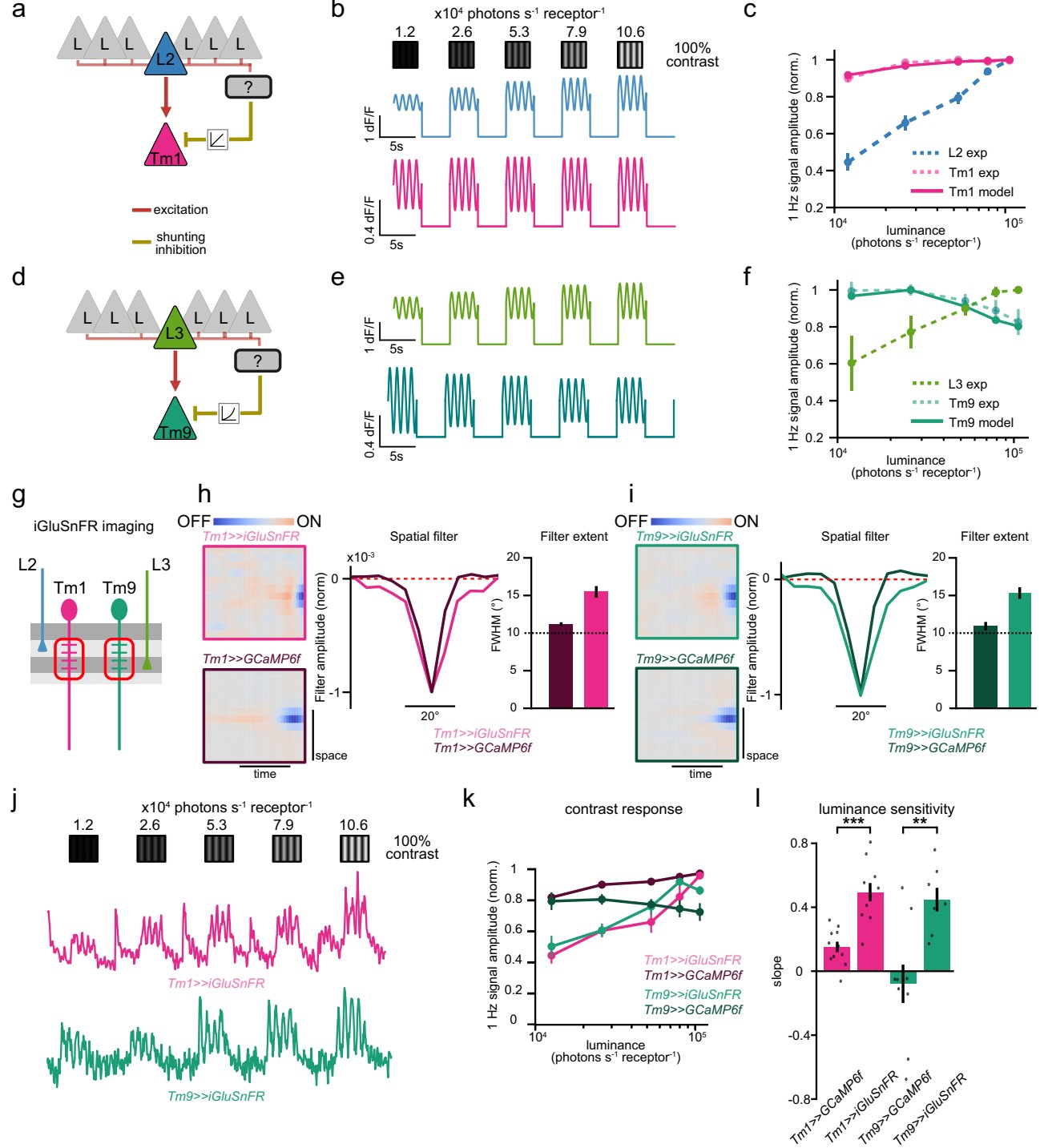

**Fig. 5 | A normalization model using shunting inhibition explains luminance gain control in medulla neurons. a** Sketch of the proposed circuit implementing normalization (luminance gain control) via spatial pooling and shunting inhibition. The Tm neuron's main LMC input is normalized by the input of a wide-field neuron via shunting inhibition. Tm1 circuit with the main L2 presynaptic input and a linear normalization factor. **b** Simulated L2 (blue) and Tm1 (magenta) traces based on the normalization model in (**a**). **c** Amplitude of contrast responses (F1 amplitude) across luminance from the simulated Tm1 responses (magenta, solid lines) and experimental data from Tm1 (magenta, dashed) and L2 (blue, dashed) neurons. Experimental data from Figs. 1, 2. **d** Tm9 circuit with the main L3 presynaptic input and a non-linear normalization factor. **e** Same as (**b**) for L3 (green) and Tm9 (teal). **f** Same as (**c**) for Tm9 and L3. Simulated Tm9 responses (teal, solid lines) and experimental data from Tm9 (teal, dashed) and L3 (green, dashed). **g** Imaging the glutamate input onto Tm1 and Tm9 dendrites using iGluSnFR. **h** Ternary white-noise stripes to extract spatio-temporal receptive fields (STRF) via reverse correlation analysis. Left: STRF of Tm1 dendritic glutamate (magenta frame) and Tm1 calcium at the axon terminals (dark magenta frame). Color axis depicts positive (ON)−negative (OFF) correlation with the stimulus. Middle: Spatial filters of STRFs.

Right: Quantification of the FWHM of the spatial filters. **i** Same as (**h**) for Tm9 neurons. **j** iGluSnFR responses of a single Tm1 (magenta) and Tm9 (teal) neurons to drifting 1 Hz gratings of constant 100% Michelson contrast and changing luminances. **k** Mean normalized contrast responses (F1 amplitude) of Tm1 dendritic glutamate (magenta) Tm9 dendritic glutamate (teal) signals. Tm1 and Tm9 calcium signals at the axon terminals (dark teal) for the same stimuli are shown for comparison (from Fig. 2). **l** Slopes of the contrast responses depicting the log-luminance dependence for each genotype. **$p < 0.01$, ***$p < 0.001$, two-sided Student's *t*-test. *P* values: Tm1»GCaMP6f, Tm1»iGluSnFR = 7.2124e-05, Tm9»GCaMP6f, Tm9»iGluSnFR = 0.0056. Error bars represent ±SEM. Means are calculated across cells for (**h**, **i**) and across flies for (**k**, **l**). Model values in (**c**, **f**) come from 1000 simulated traces, using sinusoidal stimuli with a random phase to simulate different spatial locations. Sample sizes are given as # cells for (**h**, **i**) and # flies(cells) for (**c**, **f**, **k**, **l**). For (**c**, **f**), L2 exp *n* = 19(141), Tm1 exp *n* = 11(148), L3 exp *n* = 8(150), Tm9 exp *n* = 9(111). For (**h**, **i**), Tm1»iGluSnFR *n* = 23 cells, Tm1»GCaMP6f *n* = 151 cells, Tm9»iGluSnFR *n* = 39 cells, Tm9»GCaMP6f *n* = 53 cells. For (**j**–**l**), Tm1»GCaMP6f: *n* = 11(148), Tm1»iGluSnFR: *n* = 10(78), Tm9»GCaMP6f: *n* = 9(111), Tm9»iGluSnFR: *n* = 7(23).

---

quadratic function of the pooled L3 responses[40]. Overall, divisive normalization based on shunting inhibition using a circuit that pools luminance signals via wide-field neurons explains contrast responses in dynamic luminance conditions, but with cell-type specific differences in the normalization function (Fig. 5a–f).

## Tm1 and Tm9 neurons receive wide glutamatergic inputs

Next, we investigated the biophysical mechanisms that implement luminance gain control. Tm1 and Tm9 neurons receive cholinergic input from their major columnar inputs, L2 and L3[41]. Additionally, spatial pooling requires information from several columns of visual processing to converge onto a single neuron. In the medulla, Distal medulla (Dm) neuron processes span multiple columns and thus are candidates for implementing spatial pooling[32]. The majority of Dm neurons exhibit a glutamatergic phenotype[41]. We thus assessed if Tm1 and Tm9 receive glutamatergic input by expressing the glutamate sensor iGluSnFR[42] specifically in Tm1 and Tm9 neurons. The glutamate signals onto both Tm1 and Tm9 dendrites increased to the onset of full field OFF stimuli and decreased in response to ON stimuli (Supplementary Fig. 4a).

Next, we used white noise stripes and reverse correlation analysis to extract the spatio-temporal RFs of the dendritic glutamate signal (Fig. 5g–i). For comparison, we also measured the neurons' RFs using GCaMP6f. Whereas the spatial filters extracted as the full-width-half-maximum (FWHM) from the calcium signal in Tm1 and Tm9 axon terminals were around 10°, the spatial signals of their dendritic glutamate inputs measured around 15°. This is within the spatial range over which luminance modulated those neurons' responses (Fig. 4), and wider than the calcium-based filters (Fig. 5g–i). This indicates that glutamatergic inputs provide multi-columnar information, making them candidates for implementing luminance gain control. We then probed the luminance gain control properties of the incoming glutamate signal using sinusoidal gratings of constant contrast and changing luminances (Fig. 5j). Unlike the calcium signals recorded in Tm1 and Tm9, the glutamate signals onto their dendrites scaled with luminance (Fig. 5k, I). These results predict that a glutamatergic signal transformation implements the luminance gain control on the Tm neuron dendrites.

## GluClα is required for luminance gain control in Tm9

A candidate glutamatergic mechanism to implement luminance gain control via shunting normalization could be the glutamate-gated chloride channel, GluClα[43]. To test this, we cell-type specifically knocked out *GluClα* using FlpStop. Here, an inverted STOP cassette is cell-type specifically flipped into the 'disrupting' orientation that interferes with transcription and translation (Fig. 6a)[44]. A tdTomato

marker present in the STOP cassette confirmed that the inversion happened in all Tm9s and Tm1s (Fig. 6b, Supplementary Fig. 5a). Loss of *GluClα* in Tm1 did not change the luminance gain control characteristics of the responses (Supplementary Fig. 5b–d). Loss of *GluClα* in Tm9 neurons resulted in responses that were positively correlated with luminance (Fig. 6c, d), similar to LMCs (Fig. 1). Responses of Tm9 neurons lacking *GluClα* depended significantly more on luminance than both genetic controls demonstrating that GluClα is required for stable contrast responses in Tm9 (Fig. 6e). Thus, in line with the finding that distinct computations explain luminance gain control in Tm1 and Tm9, they also differ in their biophysical implementation with GluClα being a molecular component implementing normalization for luminance gain control in Tm9.

## Dm12 is a presynaptic candidate for implementing luminance gain control in Tm9 neurons

Spatial pooling and glutamatergic inhibition are involved in luminance gain control in Tm9. The relevant circuit element that can implement this should be a glutamatergic wide-field neuron that inhibits Tm9, and that does not compartmentalize the signal but responds to wider regions of visual space. Tm9 has been shown to have variable pre-synaptic connectivity in different columns of the eye[45], so this pooling task might be distributed across different cell types. To identify neurons that can contribute to the task we used the Flywire connectome of a full adult fly brain (FAFB), the first to include a large enough portion of the optic lobes to cover wide-field neurons[45–48]. One of the major inputs of Tm9 is the distal medulla neuron Dm12, which is glutamatergic based on connectomics and transcriptomics data (Fig. 7a)[41,49]. Dm12 neurites span three medulla columns, corresponding to ~15° of visual space, making it a prime candidate for spatial pooling (Fig. 7b). In each column, Dm12 neurons receive major inputs from L3 neurons[50]. To probe functional connectivity, we expressed csChrimson[51] in Dm12, while recording GCaMP6f signals in Tm9 neurons (Fig. 7c). We used a red LED to activate csChrimson and recorded in blind flies to avoid confounding effects of light stimulation (Fig. 7c). Activating Dm12 decreased the calcium signals in Tm9 showing that they are functionally connected and that Dm12 provides inhibitory input to Tm9. Next, we measured the visual response properties of Dm12. In response to full-field flashes, Dm12 calcium signals increased for OFF and decreased for ON stimuli (Fig. 7d).

Next, we asked how local or global Dm12 responded to visual inputs and characterized the response properties of single Dm12 neurites using calcium imaging. Mapping their RF using vertical bars showed that single columnar projections of Dm12 neurons have wide RFs with a mean FWHM of ~17° (Fig. 7e–g). This confirms that Dm12 neurons integrate information from multiple visual columns. In

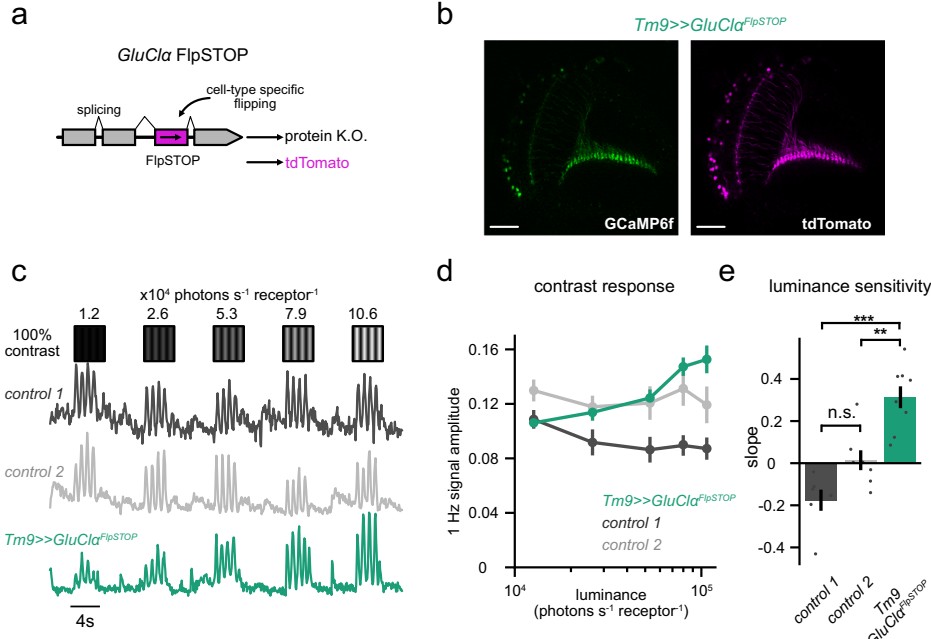

**Fig. 6 | GluClα is required for luminance gain control in Tm9. a** Cell-type specific disruption of *GluClα* using FlpSTOP where a non-disrupting STOP cassette is inverted via cell-type specific expression of Flp recombinase, visualized by the expression of tdTomato. **b** Two-photon micrograph, visualizing GCaMP6f (green) and tdTomato expression (magenta) in all Tm9 neurons, confirming cell-type specific flipping of the STOP cassette. Scale bars are 50 μm. **c** Calcium signals of single axon terminals of *Tm9»GluClα^FlpSTOP* (teal) and control genotypes (black: no Flp, gray: heterozygous control) in response to drifting 1 Hz gratings of constant 100% Michelson contrast and changing luminances. **d** Mean contrast responses (F1 amplitude) of each genotype across luminance. **e** Slopes of the contrast responses depicting the log-luminance dependence for each neuron. **p < 0.01, ***p < 0.001, one way ANOVA with post-hoc Tukey HSD test. *P* values for pairwise comparisons: control 1–control 2 = 0.0709, control 1 - *Tm9»GluClα^FlpSTOP* = 1.7820e-05, control 2 - *Tm9»GluClα^FlpSTOP* = 0.0022. Error bars represent ±SEM. Means are calculated across flies. Sample sizes for (**d**, **e**), control 1: *n* = 6(37), control 2: *n* = 7(68), *Tm9»GluClα^FlpSTOP*: *n* = 8(92). Sample sizes are given as: flies(cells).

summary, our data reveals that the glutamatergic Dm12 neuron is a wide-field neuron that pools luminance information from many L3 neurons and inhibits Tm9 neurons. Last we tested if Dm12 is the sole contributor of luminance gain in Tm9 neurons and silenced Dm12 neurons. Expressing the inwardly-rectifying potassium channel Kir2.1[52] in Dm12 neurons did not affect luminance gain of the Tm9 cell population suggesting that luminance gain control is a distributed task and that further Tm9 input neurons are involved in implementing luminance gain (Supplementary Fig. 6).

Overall, a computational analysis of natural scenes, a physiological characterization of specific Tm and Dm cell types, connectomics analysis, and circuit models converge on the idea that local spatial pooling of luminance and a correction of contrast by this luminance signal ensures the stable processing of contrast in dynamically changing environments.

## Discussion

Rapid luminance gain control is crucial for vision since it ensures stable contrast representations upon fast changes in the environment. Here, we revealed its algorithmic, cellular, and biophysical basis in the *Drosophila* visual system. Stable contrast estimation is first implemented in specific third-order visual neurons. Tm1 exhibits luminance-invariant contrast responses whereas Tm9 enhances contrasts in low contextual luminance. Both neurons implement luminance gain control through local spatial pooling to scale contrast distributions with local scene luminance, which is relevant when viewing or navigating natural scenes. A normalization model that combines spatial pooling and shunting inhibition explains the emergence of luminance gain control in visual circuitry. Accordingly, both neurons receive wide-field glutamatergic signals onto their dendrites and Tm9 neurons require the inhibitory channel GluClα to implement luminance gain control. Taken together, our work advances our understanding of how visually

guided animals can perform stable contrast processing in challenging, natural environments.

Stable fly behavior arises from interactions between contrast-sensitive and luminance-sensitive information[6,10]. While LMCs do not maintain luminance-invariant responses in rapidly changing conditions, the dendrites of downstream medulla neurons Tm1 and Tm9 are the sites where rapid luminance gain control is implemented. The postsynaptic direction-selective T5 neurons receive their major inputs from Tm1, Tm2, Tm4 and Tm9 neurons[35,36] and maintain luminance-invariant signal. Thus, the luminance-invariant signal in T5 likely originates in Tm1 and Tm9 neurons and their input onto distinct dendritic regions of T5 neurons prevents luminance of the visual scenes from becoming a confounding variable in motion computation. In line with this, downstream LPTCs, wide-field neurons that sample many local motion cues, also show luminance-invariant responses in recordings done in bigger flies[53,54]. Our experiments were conducted within a luminance range that aligns with the peak activity period of fruit flies during the day[55,56]. The luminance range encountered in natural scenes throughout the day is wider[1]. Previous research showed that flies display luminance-invariant behavior across several orders of magnitude. This depends on post-receptor luminance gain control, which is mediated by the luminance-sensitive L3 neuron at all luminance regimes[6,11]. Thus, the luminance gain control circuits and mechanisms identified here are likely applicable across the extensive range of luminance conditions found in natural settings.

Why is luminance gain control implemented in the medulla rather than in photoreceptors or LMCs? First, background luminance must be accurately and rapidly estimated. Photoreceptors, which are highly susceptible to noise, will distort contrast representations under rapidly changing conditions[57]. Post-receptor locations benefit from mechanisms aimed to reduce noise like synaptic and spatial pooling[8,58,59]. The medulla contains many horizontally projecting cell

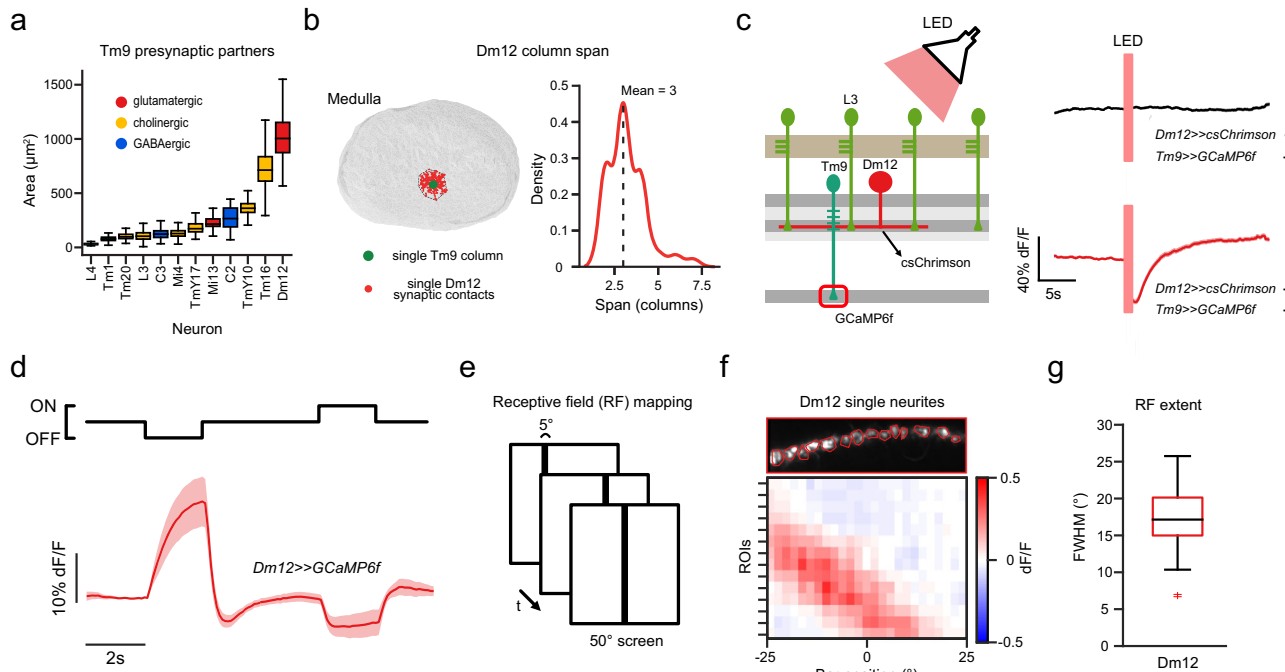

**Fig. 7 | Dm12 can implement luminance gain control in Tm9 neurons. a** Area that the neurites of Tm9 presynaptic partners in the medulla cover. Data from the FAFB dataset[46–48]. **b** Left: EM reconstruction of medulla with neurites of a single Dm12 neuron (red) and a single Tm9 neuron (green). Right: Distribution of Dm12 column span. **c** Calcium imaging of Tm9 neurons while optogenetically activating Dm12 neurons expressing csChrimson using a red LED. Stimulation of flies expressing csChrimson in Dm12 (red) and control flies without csChrimson expression (black). **d** In vivo two-photon calcium imaging of Dm12 neurites (red) while presenting full field ON and OFF flashes. **e** Mapping of the receptive field of Dm12 neurons using 5°

static OFF bars. **f** Calcium responses of single Dm12 neurites of a single fly to OFF bars across bar positions. **g** Full-width-at-half-maximum (FWHM) of the receptive fields of Dm12 single neurites. Sample sizes are given as # cells in (**a**): Tm9: $n = 700$ neurons. For (**b**): Dm12 $n = 118$ neurons. Sample sizes are given as # flies(cells) in (**c**): *Tm9»GCaMP6f* $n = 6(304)$, *Dm12»csChrimson; Tm9»GCaMP6f* $n = 8(310)$. (**d**): $n = 13(261)$. (**g**): $n = 6(60)$. Boxplots in (**a, g**) are defined as, center: median, bounds of box: 25% and 75% percentiles, maxima and minima: the most extreme data points. Error bars in (**c, d**) represent ±SEM. Means are calculated across flies.

types like Dm and Pm neurons that can spatially pool signals. Horizontal connectivity in the lamina exists as well but primarily focuses on creating antagonistic center-surround structures for contrast signal extraction and sharpening[58]. At the same time, not all third-order neurons exhibit luminance gain-corrected signals, implying that certain pathways are dedicated to other computations. It may thus be advantageous to implement luminance gain control in some of the many parallel medulla pathways, rather than the few LMCs downstream of photoreceptors. In vertebrates, rapid luminance gain control might emerge at a similar stage of visual processing. Vertebrate photoreceptors and bipolar cells have slower adaptation mechanisms and do not exhibit rapid gain control[60,61]. However, fourth-order neurons in the LGN exhibit near-instantaneous luminance gain control[2]. In between, some of the RGCs show a luminance gain control mechanism operating in dim light[59], yet it is not clear if this generalizes to brighter conditions, and to rapid changes occurring within natural scenes.

All visually guided animals encounter rapid luminance changes whenever navigating through their environments. Natural scene analysis showed that, in the absence of corrective mechanisms, contrast computation is challenged. Normalizing contrast responses using local spatial pooling achieves stable contrast encoding. Consistent with a need for local luminance gain control, humans have luminance-invariant perception in spatially non-uniform illumination[62]. The extent of spatial pooling plays a pivotal role in ensuring a reliable and accurate estimation of luminance. On one hand, luminance correlations in natural scenes drop rapidly with distance[2,3], explaining why too wide spatial pooling leads to inaccurate luminance background estimations and to poor contrast representations. On the other hand, too local pooling heavily modifies the contrast distribution structure, compromising the neurons' ability to resolve structure in a scene.

Overall, local pooling extents that avoid such extreme conditions optimize luminance background estimation, ensuring reliable contrast representations of visual cues in typical, inhomogeneous natural scenes.

The demand for local spatial pooling when viewing natural scenes is reflected in the physiological properties of Tm1 and Tm9: their responses are modulated by luminance within a range of 8°–20°, and the RF of a neurite of the major wide-field input to Tm9, Dm12 spans ~15°, arguing that this circuit evolved to match the demands of natural visual scenes. The response modulation is likely achieved by pooling input from several luminance-sensitive L3 neurons, Dm12's major input. In line with this idea, Dm12 response dynamics closely match the sustained, luminance-sensitive L3 responses[19,63]. However, Dm12 only provides input to Tm9 in 45% of columns[45] and silencing Dm12 alone does not affect Tm9 luminance gain control, arguing that other neurons contribute to luminance gain control in other columns. This is interesting, because other wide-field neurons could act across other spatial scales. Such a heterogeneity could ensure that different spatial scales present in natural scenes are accommodated across the fly eye. While natural environments are shared by different animals, species-specific constraints, such as differences in the scene statistics, optical apparatus or in the animal's behavioral repertoire, might further tune the mechanisms to achieve reliable contrast representations. For example, species living in thick rainforests encounter natural scenes with low coherence and high contrast energy, whereas species living in open spaces like the savannah will encounter scenes with high coherence and less contrast energy[64]. Low coherence might require narrower sampling of luminance values, which are changing more rapidly across space. Yet, the main computational principles likely remain valid across visual systems. Certainly, circuit substrates for spatial

pooling are present across levels of hierarchies in the vertebrate visual system as well[65].

Contrast of an object is computed by dividing the difference between object and its background by local background luminance. Many lines of evidence argue that this is achieved by an inhibitory divisive normalization, a common algorithm in the brain[12]. One of its major purposes is to keep feature representations stable, such as the concentration-invariant recognition of odors in flies[13,14], the contrast-invariant encoding of image patterns in visual cortex[15] or the invariant representation of objects in the ventral visual pathway[66]. In the fly visual system, where the normalization mechanism ensures a stable representation of contrast, Tm1 and Tm9 dendrites both receive major input from columnar, cholinergic LMCs, and additionally incorporate wide-field inputs, which are either glutamatergic or GABAergic[41,45]. Tm1 and Tm9 also both express various acetylcholine, GABA, and glutamate receptors[41,67,68]. A picture emerges in which the columnar cholinergic input is normalized by the inhibitory wide-field inputs. This leads to the division of the center contrast value with the local background luminance value. Modeling and experimental results converge on a scenario in which this normalization step is achieved via shunting inhibition using a spatially pooled signal. Unlike previously described normalization processes in the olfactory system which are based on GABAergic inhibition[13,69], Tm9 neurons use the glutamate-gated chloride channel GluClα which was shown to mediate shunting inhibition in other visual system neurons[43]. Divisive normalization in the fly brain can thus emerge via diverse inhibitory mechanisms. Recently, a computational survey of the fly visual system based on EM connectivity argued that Dm-mediated normalization is a wide-spread phenomena and serves specific functions such as normalization over space or features[70]. In the visual cortex there are several mechanisms proposed to explain normalization but so far, no molecular component has been causally identified. There is evidence against a GABAergic implementation[71] such that non-GABAergic inhibitory mechanisms might also be involved in vertebrate visual systems.

Two of the four major inputs to the direction-selective T5 cells implement luminance gain control, but with distinct characteristics: Luminance gain control in one channel (Tm1) ensures luminance-invariant contrast representations. The other channel (Tm9) instead amplifies contrast in low contextual luminance. Enhancing contrast in low luminance might be advantageous because contrast perception in dim light is challenging due to low signal-to-noise ratios. At the same time, predators often use dim light as a strategy for prey capture, and dark regions dominate in natural scenes[72,73]. The source of this enhancement in low luminance for Tm9 neurons could originate from a non-linearity provided by GluClα which was shown to non-linearly enhance motion responses in downstream neurons[43]. Furthermore, Tm9 receives major inputs from L3 neurons[34,35,45], which non-linearly amplify responses to low luminances[6,10,19], while other Tm neurons receive major input from L2[34,35]. Similar to the visual system, the fly olfactory system has multiple distinct gain mechanisms, which interact in various ways[74]. Medulla neurons also exhibit complex inter-connectivity and this connectivity could provide ways for the Tm1 and Tm9 gain mechanisms to interact depending on the state of the animal and environment.

The other two of the four major OFF medulla neurons converging onto T5 dendrites, Tm2 and Tm4, show luminance-dependent representations of contrast. Both Tm2 and Tm4 neurons exhibit contrast gain, suggesting that the medulla circuitry processes fluctuations in contrast and luminance independently[75,76]. This raises the question of how parallel inputs from neurons with luminance gain control (Tm1 and Tm9) and neurons with contrast gain control (Tm2 and Tm4) are integrated to ensure stable motion processing in T5 neurons. In the vertebrate LGN, luminance and contrast gain control mechanisms exist and operate independently reflecting the independence of these features in natural scenes[2]. A model incorporating two independent resistor-capacitor (RC) stages, one for luminance gain control and another for contrast gain control, can account for the observed responses[77]. These RC stages shape the linear RF of LGN neurons to maintain stable responses across varying stimuli. While the biophysical and circuit implementations may differ in the fly, the principle of independent gain control mechanisms suggests a widely used strategy for maintaining visual stability In the vertebrate LGN, luminance and contrast gain also operate independently, Taken together, a combination of parallel gain mechanisms ensures the stable extraction of downstream visual features. Both vertebrates and invertebrates are subjected to similar natural scenes. Thus, our findings likely offer widely applicable insights into how visual systems cope with rapid luminance changes encountered in natural environments.

## Methods

### Fly husbandry and genotypes
Flies were raised on molasses-based food on a 12:12 h light:dark cycle at 25 °C and 65% humidity. Parental crosses were flipped every 2–3 days onto new food. Two-photon experiments were conducted at room temperature (20 °C). Female flies 2–4 days after eclosion were used for experiments. Genotypes used in experiments are given below (Table 1). Fly lines are obtained from following sources: $L1^{c202}$-Gal4[78], $L2^{21Dhh}$-Gal4[78], $L3^{MH56}$-Gal4[79], $Tm1^{R74G01}$-Gal4[80], Tm2-split-Gal4[81], Tm4-split-Gal4[81], Dm12-split-Gal4[82], $Tm9^{24C08}$-lexA[83], $T4/T5^{R59E08}$-lexA[83], $Tm9^{R42C08}$-Gal4[80], $GluClα^{FlpStop-ND67}$.

### In vivo calcium imaging
To dissect flies for imaging, adult flies were anesthetized on ice and mounted into a stainless-steel custom-made fly holder containing a hole sized to fit the thorax of the fly. The fly head was tilted to expose the cuticle at the back of the fly's head. The left side of the head was fixed to the holder using a UV-sensitive glue (Bondic) and legs and proboscis were glued to the body using a low temperature melting wax to prevent brain motion. Fine breakable razor blades and sharp forceps were used to remove the cuticle on the right side of the head, fat bodies, and trachea. The dissection was done in saline containing 103 mM NaCl, 3 mM KCl, 5 mM TES, 1 mM NaH2PO4, 4 mM MgCl2, and 26 mM NaHCO3. Imaging was done in saline additionally containing calcium and sugars (1.5 mM CaCl2, 10 mM trehalose, 10 mM glucose, 7 mM sucrose). The imaging solution was perfused across the fly brain and carboxygenated to achieve a constant pH of 7.3.

Two photon experiments for Figs. 1, 2, 5–7, Supplementary Fig. 1, 4 were done using a Bruker Investigator microscope (Bruker, Madison, WI, USA) equipped with a 25 x/NA1.1 objective (Nikon, Minato, Japan). An excitation laser (Spectraphysics Insight DS+) was tuned to 920 nm to excite GCaMP6f. tdTomato expression in FlpStop experiments was recorded using a fixed 1040 nm laser line on the same set-up. For Fig. 4, Supplementary Fig. 3, 5 we used a Bruker Ultima microscope equipped with a 20 x/NA1.0 objective (Leica, Wetzlar, Germany), coupled to a fixed excitation laser of 930 nm (YLMO-930 Menlo Systems, Martinsried, Germany). Both setups were equipped with a SP680 shortpass filter, a 560 lpxr dichroic filter and 525/70 and 595/50 emission filters for the emitted light. Bruker PrairieView software was used to acquire the data. Images were recorded at a frame rate of 10–15 Hz and using an optical zoom of 6–8×. Typically less than 20 mW of laser power was delivered to the specimen, measured at the objective.

### Visual stimulation
For both experimental configurations, visual stimuli were presented via a DLP LightCrafter 4500 (Texas Instruments, Dallas, TX, USA) with a frame rate set to 100 Hz. To filter the stimulus a 482/18 bandpass filter and an ND1.0 neutral density filter (Thorlabs) were mounted on the projector. The coordination of stimulation and imaging was achieved through a DAQ USB-6211 device (National Instruments, Austin, TX, USA). For the Bruker Ultima setup, visual stimuli were scripted in

**Table 1 | Genotypes used for experiments**

| Name | Genotype |
|---|---|
| L1»GCaMP6f | w+; L1^{c202}Gal4/+; UAS-GCaMP6f/+ |
| L2»GCaMP6f | w+; UAS-GCaMP6f /+; L2^{21Dhh}-Gal4 /+ |
| L3»GCaMP6f | w+; L3^{MH56}-Gal4 /+; UAS-GCaMP6f /+ |
| Tm1»GCaMP6f | w+; UAS-GCaMP6f /+; R74G01-Gal4 /+ |
| Tm2»GCaMP6f | w+; R28D05-p65ADZp^{attP40} /UAS-GCaMP6f; R82F12-ZpGdbd^{attP2} /+ |
| Tm4»GCaMP6f | w+; R53C02-p65ADZp^{attP40} /UAS-GCaMP6f; R60H04-ZpGdbd^{attP2} /+ |
| Tm9»GCaMP6f | w+; Tm9^{24C08}-lexAp65^{attP40}, lexAop-GCaMP6f^{attP5} /+;+/+ |
| T4/T5»GCaMP6f | w+; R59E08-LexA^{attP40}, lexAop-GCaMP6f- p10su(Hw)^{attP5}/+; +/+ |
| Tm1»iGluSnFR | w+;+/+; R74G01-Gal4 / UAS-iGluSnFR A184AattP2 |
| Tm9»iGluSnFR | w+;+/+; GMR42C08-Gal4attP2 / UAS-iGluSnFR A184A^{attP2} |
| Tm9»GluClα^{FlpSTOP} | w+; UAS-GCaMP6f,UAS-Flp /+; GluClα^{FlpStop-ND}/GMR42C08- Gal4^{attP2}, GluClα^{MI14426} |
| Tm9 control 1 (no Flp control) | w+; UAS-GCaMP6f /+; GluClα^{FlpStop-ND}/GMR42C08- Gal4^{attP2}, GluClα^{MI14426} |
| Tm9 control 2 (heterozygous control) | w+; UAS-GCaMP6f,UAS-Flp /+; GluClα^{FlpStop-ND}/GMR42C08-Gal4^{attP2} |
| Tm1»GluClα^{FlpSTOP} | w+; UAS-GCaMP6f,UAS-Flp /+; GluClα^{FlpStop-ND}/GMR74G01- Gal4^{attP2}, GluClα^{MI14426} |
| Tm1 control 1 (no Flp control) | w+; UAS-GCaMP6f /+; GluClα^{FlpStop-ND}/GMR74G01- Gal4^{attP2}, GluClα^{MI14426} |
| Tm1 control 2 (heterozygous control) | w+; UAS-GCaMP6f,UAS-Flp /+; GluClα^{FlpStop-ND}/GMR74G01-Gal4^{attP2} |
| Tm9»GCaMP6f Dm12»csChrimson | w+, norpA36/ Y; Tm9^{24C08}-lexAp65^{attP40}, lexAop-GCaMP6f^{attp5} /SS00359(Dm12-splitGAL4-AD); UAS-CsChrimson::mVenus^{attP2}/ SS00359(Dm12-splitGAL4-DBD) |
| Dm12»UAS-GCaMP6f | w+; SS00359 (Dm12-splitGal4-AD); SS00359 (Dm12-splitGal4-DBD)/UAS-GCaMP6f |
| Tm9»GCaMP6f Dm12»UAS-Kir2.1 | w+; Tm9^{24c08}-lexAp65^{attP40}, lexAop-GCaMP6f^{attp5} /SS00359(Dm12-splitGAL4-AD); UAS-Kir2.1/SS00359(Dm12-splitGAL4-DBD) |

Python[84] based on the PsychoPy package[85]. The projection was directed onto a 9 × 9 cm rear projection screen positioned at a 45° angle relative to the fly, covering 80° of visual space in both azimuth and elevation. The Bruker Investigator setup employed custom-written C++ and OpenGL software[18] for stimulus generation, with the stimuli displayed on an 8 × 8 cm rear projection screen encompassing 60° of visual space in both azimuth and elevation. The maximum luminance value (Imax) recorded at the fly's position was $2.17*10^5$ photons s$^{-1}$ photoreceptor$^{-1}$ for the Bruker Investigator and $2.4*10^5$ photons s$^{-1}$ photoreceptor$^{-1}$ for the Bruker Ultima setup.

**Drifting sinusoidal gratings.** Full field drifting sinusoidal gratings moved at 1 Hz temporal frequency (speed: 30°s $^{-1}$, spatial wavelength: 30°) and maintained 100% Michelson contrast at varying mean luminance levels of 1.2, 2.6, 5.3, 7.9, and 10.6 $*10^4$ photons s$^{-1}$ photoreceptor$^{-1}$. Each drifting epoch lasted for 4 s, presented in a randomized sequence, followed by 4 s of full-field mean luminance. Each epoch was repeated at least three times. When recording T4/T5 neurons, the gratings moved in two opposing horizontal directions, while for other neurons, only a single direction was shown to the fly. As an additional stimulus, we used different Michelson contrasts of 20, 40, 60, 80, and 100%, all interspersed with the previously mentioned mean luminance values.

**OFF moving edges.** A 1 s full field presentation of a given luminance was followed by a moving dark edge at a constant 100% Weber contrast $(I_{edge} - I_{background})/I_{background}$ moving at 30°s$^{-1}$. The full field luminance before the OFF edges were 1.2, 2.6, 5.3, 10.6, 16, 21.4 $*10^4$ photons s$^{-1}$ photoreceptor$^{-1}$. The edge epochs were randomized and interleaved by 4 s of darkness.

**Ternary white noise of squares for online RF mapping.** The white noise frames comprised 2.5° squares measuring covering the entire screen. Each square changed its brightness at a rate of 20 Hz,

transitioning between full darkness, intermediate brightness ($0.5*I_{max}$), or full brightness ($I_{max}$) with equal probability. Preceding the white noise presentation, a 4 s interval of full-field intermediate brightness was provided to capture the baseline signal of neurons. The stimulus was presented for ~3–4 min, allowing for the extraction of a receptive field (RF) with a robust signal-to-noise ratio to estimate the RF centers.

**Drifting gratings with various sizes for online RF mapping and stimulation.** Drifting gratings of 100% Michelson contrast and different luminances were presented as described above but using circular grating shapes of varying sizes. The diameter ranged from 5° to 30° with 5° increments. The grating was placed at the center of the mapped RF. Each grating epoch lasted for 4 s, followed by 4 s of full-field mean luminance presented in a randomized sequence. The epochs were repeated at least three times.

**Constant grating with background annulus for online RF mapping and stimulation.** Drifting grating of 100% Michelson contrast and a mean luminance of $5.3*10^4$ photons s$^{-1}$ photoreceptor$^{-1}$ were presented as above, but at a fixed 5°diameter of the drifting grating. Then we added an annulus (background annulus) with an inner circle diameter of 5° so that the drifting grating was visible. The outer circle diameter ranged from 7.5° to 27.5° with 2.5° increments. The luminance values for the annuli were uniformly set at 1.2, 2.6, 7.9, and 10.6 $*10^4$ photons s$^{-1}$ photoreceptor$^{-1}$. Between epochs, a static 5° grating with a mean luminance of $5.3*10^4$ photons s$^{-1}$ photoreceptor$^{-1}$ was interleaved. Each epoch lasted 3 s with an interleave duration of 3 s. The epochs were randomized and repeated three times.

**Periodic full-field flashes.** The stimulus consisted of two epochs of full-field 100% Weber contrast ON ($I_{max}$) and OFF flashes lasting 5 s. The epochs were repeated at least 8 times.

**Ternary white noise stripes.** The white noise frames were composed of vertical stripes, each with a width of 5°, covering the whole screen. Every stripe modulated its brightness at a frequency of 20 Hz, transitioning between full darkness, intermediate brightness ($0.5 \ast I_{max}$), or full brightness ($I_{max}$) with equal probability. Prior to the white noise sequence, a 4 s interval of full-field intermediate brightness was introduced to capture the baseline signal of neurons. The stimulus continued for ~10 min.

**Static OFF stripes.** Stimuli consisted of 5° vertical bars with 100% Michelson contrast, presented on a bright background. The individual bar positions covered the screen with 2° shifts. In each trial, the positions of the bars were randomized. A single bar was briefly presented at each position for 1 s, followed by a 1 s inter-stimulus interval during which the background was displayed. The epochs were presented four to five times each.

## Optogenetics

A solution of 100 mM all-trans-retinal (ATR) in ethanol (EtOH) was prepared by diluting 25 mg of ATR in 878.92 µl EtOH. Similar to the approach outlined in ref. 86, the food was enriched with the ATR solution to achieve a final concentration of 1 mM. Handling of ATR and preparation of food enriched with ATR were done in darkness to avoid ATR degradation. The flies were placed on the ATR-supplemented food after eclosion, and the vials were covered with aluminum foil to prevent light exposure. Vials containing flies on ATR food were then placed in incubators set at 25 °C for a minimum of 3 days before commencing the experiments.

A 625 nm LED (Thorlabs) delivered light through the objective to activate csChrimson for 625 ms flickering at 40 Hz. Each pulse lasted 5 ms and had equal power of 29.40 µW mm$^{-2}$. Five trains per recording were repeated every 30 s.

## Data analysis

Data processing was in general done using Python 2.7. Motion correction was done via the Hidden Markov Model implemented in the SIMA Python package[87]. Only for Dm12 optogenetics and imaging datasets, data processing was done using MATLAB (The MathWorks Inc, Natick, MA). Image registration was conducted to correct motion artifacts and align multiple time series corresponding to different stimuli. Image series underwent alignment through rigid registration, involving the translation of images to maximize the cross-correlation with a designated template frame. A template image was generated by calculating the signal-to-noise ratio (SNR) for the initial 300 frames of the time series (or the complete time series if shorter) and constructing the template by averaging frames up to the onset of the SNR plateau.

In both processing pipelines, the selection of regions of interest (ROIs) was conducted through either manual selection or utilizing the spatiotemporal Independent Component Analysis (ICA) segmentation algorithm within the SIMA package (for T4/T5 axon terminals). The time traces corresponding to each ROI were obtained by averaging the signal within the designated area, followed by background subtraction. To calculate relative changes in signal (ΔF/F0), the mean of the trace served as the baseline (F0) if not indicated otherwise in the following sections. Subsequently, trial averaging and analyses specific to the stimuli were carried out. To filter out noisy ROIs, we calculated the correlation between individual trials and averaged them to obtain a reliability value. We filtered out all ROIs with a reliability value smaller than 0.6, decided upon visual inspection, unless stated otherwise. Stimulus-specific analyses were performed individually for each ROI and later averaged first within, and then across flies.

**Dm12 imaging.** After trial averaging, ROIs were filtered based on a response quality index as in Ramos-Traslosheros and Silies[22] using a threshold of 0.5. Dm12 single neurite spatial RFs were extracted and a single gaussian was fit to extract the FWHM following the protocol in Ramos-Traslosheros and Silies[22].

**Drifting gratings.** As the drifting gratings had a specific temporal frequency (1 Hz), contrast responses were determined by calculating the signal amplitude at the grating temporal frequency through a Fourier transformation. In some cases, for ease of comparison regarding luminance dependency across different neuron types, the contrast responses were normalized by the maximum value within each fly (mentioned in the y axis of the quantification plots). Additionally, luminance dependency was assessed by fitting a linear regression line to the contrast response versus log-luminance values and extracting the slope.

The generation of contrast-luminance heat maps involved linear spline interpolation utilizing the SciPy function scipy.ndimage.zoom() on the original data (mean across flies) derived from grating contrast responses for 5 contrasts and 5 luminances. The original data constituted a $5 \times 5$ matrix representing amplitudes for each contrast-luminance pair. A zoom value of 5 was applied, increasing the resolution from $5 \times 5$ to $25 \times 25$, with an interpolation order of 1 (linear). Subsequently, the interpolated maps were smoothed using a Gaussian filter with a sigma of 10 before 8 contour lines were plotted. This methodology enhances the visualization of the contrast-luminance space, facilitating an exploration of how responses vary with contrast and luminance.

**OFF moving edges.** For each epoch, the edge responses of ROIs were computed as the difference between the mean luminance during the preceding 1 s presentation and the peak response triggered by the OFF edge. As in the drifting gratings analysis, luminance dependency was assessed by fitting a linear regression line to the edge response versus log-luminance values and extracting the slope.

**Reverse correlation analysis for the white noise stimulation.** Initially, the time traces of ROIs were interpolated to 20 Hz to match the update rate of the white noise stimulation paradigm. Relative changes in signal (ΔF/F0) were then computed using the mean of the first 4 s of full-field intermediate brightness stimulation as F0. The trace was centered around the mean, aligning with the stimulus centered around 0. Subsequently, reverse correlation analysis was employed to extract the Spatiotemporal Receptive Fields (STRFs). A backward sliding window of 2 s was moved through the stimulus, and for each iteration, the stimulus values were adjusted based on the neuron's response at the beginning of the window ($r_t$). With T representing the total time of stimulus presentation, τ denoting the stimulus time window, rt representing cell response at time point t, and s(t − τ) indicating stimulus values during the time window, the equation for the STRF is:

$$STRF = \frac{1}{T - \tau} \sum_{t=\tau}^{T} r_t s(t - \tau) \tag{1}$$

To filter out noisy STRFs we used an absolute filter amplitude threshold of 0.005, decided upon visual inspection. Since iGluSnFR Tm1 recordings were noisier than other experiments, the threshold was reduced to 0.003. To further eliminate noisy STRFs in iGluSnFR Tm1 recordings, we used a SNR threshold of 10 where SNR is determined as the absolute maximum value of the STRF divided by the mean of the STRF. These thresholds were determined upon visual inspection of STRFs. Temporal filters were extracted by picking the time trace of the spatial location where the filter produced its minimum response. Spatial filters were extracted by picking the spatial trace of temporal location with the minimum response. The minimum was used since STRF amplitudes of OFF cells are negative. To calculate

the full-width-at-half-maximum (FWHM) we fitted a single Gaussian:

$$f(x) = a * e^{-\frac{(x-\mu)^2}{2\sigma^2}} \qquad (2)$$

where $\mu$ is the mean and $\sigma$ is the standard deviation. FWHM was then calculated as $2.355 * \sigma$.

**Online RF mapping.** First, the neurons' STRFs were extracted using reverse correlation analysis (described above). To get the center location of the STRFs, a two dimensional Gaussian was fitted to the spatial RF at the time where minimum filter amplitude was reached (minimum since neurons recorded are OFF cells). A two dimensional Gaussian is defined as:

$$f(x,y) = h * e^{-\frac{\left(\frac{center_x - x}{width_x}\right)^2 + \left(\frac{center_y - y}{width_y}\right)^2}{2}} \qquad (3)$$

$center_x$ and $center_y$ values were then used to place the drifting grating center in the stimulus screen. An axon terminal with a RF close to the center of the screen was selected for further stimulation. Further analysis for extracting the contrast response of the neurons to the centered drifting gratings was the same as the above-mentioned analysis for drifting gratings.

## Modeling
**L3 and L2 contrast responses.** We implemented an effective single-compartment voltage-membrane-potential model[88] to fit L3 responses to moving gratings as,

$$\frac{dv}{dt} = -(v - v_L) + g^{-1} s(t) \qquad (4)$$

Where $v_L$ represents the leakage voltage and $g$ represents the conductance corresponding to the input sinusoidal signal, $s(t)$. We use the pseudo-stationary solution of the membrane potential, where we consider the neuron time integration to be faster than the experimental time. We then implement a ramp rectifying function, $R[\cdot]$, to model the calcium response of L3 neurons as,

$$R^2(v) = \alpha R\left(v_L + g^{-1} s(t)\right)^2 \qquad (5)$$

We set the parameters of our effective model to fit the experimental calcium contrast responses of L3 neurons in response to 1 Hz drifting gratings (Fig. 1), $g^{-1} = 0.05, v_L = 0.3$ and $\alpha = 4$. Similarly, we fit the experimental parameters that reproduce L2 contrast responses.

**Contrast sampling from natural scenes.** We estimated L3 responses to the natural scene stimuli using our effective single-compartment voltage-membrane-potential model. To sample contrast responses under dynamic conditions, we generated random forward-wave trajectories following the equation,

$$T(z) = 30 \sin\left(2\pi 3\omega_z z\right) + \gamma z + 60 \mathcal{N}(0,1) \qquad (6)$$

with $\omega_z$ being the inverse of the spatial length of the natural scene and $\gamma$ an orientation weight set to $\gamma = 1/3$ for the trajectories in Fig. 3c. We compute the distribution of contrast responses from these trajectory points using a Gaussian Kernel density estimator.

We quantified how distant the contrast distributions are from different luminance conditions with the Wasserstein distance, $\mathcal{D}[\cdot]$[89]. Specifically, we defined the loss function,

$$\mathcal{L}^2 = \mathcal{D}\left[\rho(c)_S | \rho(c)_D\right]^2 + \mathcal{D}\left[P(c - \bar{c})_D | P(c - \bar{c})_{D-norm}\right]^2 \qquad (7)$$

with $\rho(c)_S$ and $\rho(c)_D$ the contrast distributions in sunny and shade conditions respectively and $\rho(c - \bar{c})_{D-norm}$ the normalized contrast distribution in shade condition. The first term in the loss function corresponds to the distance between the contrast distributions from sun and shade conditions. The second term corresponds to the distance between the shifted zero-mean distribution in sun conditions and the shifted zero-mean normalized distribution. The first term penalizes differences between luminance conditions while the second term penalizes narrow distributions as compared to the non-normalized distribution, leading to poor contrast information.

**Tm1 and Tm9 contrast responses.** We implemented a normalization model based on shunting inhibition[40], where the neuronal response, in the pseudo-stationary regime, was determined by the main input current and divided by a normalizing function of the pooling input, that is,

$$R = \frac{I(t)}{g_l + J(t)^p} \qquad (8)$$

With $I(t)$ being the main input current, $J(t)$ the pooling input, $g_l$ a leakage conductance constant and $p$ an integer. We consider the pooling input, $J(t)$, to be the sum of lamina neuron responses within a spatial region, $\mathcal{R}$, determined by the pooling size $r$, that is,

$$J(t) = \sum_{i \in \mathcal{R}} I_i \qquad (9)$$

We modeled Tm1 responses with a linear function of the pooling input, $p = 1$, and Tm9 responses with a non-linear function described by a quadratic exponent, $p = 2$.

## Analysis of FAFB electron microscopy datasets
We acquired connectomics data from the right optic lobe of the full adult female fly brain (FAFB) electron microscopic dataset using a custom-written code in Python 3.9 with the fafbseg Python package[47,48].

## Presynaptic Partner and Neurotransmitter Identity
For a total of 700 Tm9s (medulla columns), we identified ~80% of all inputs by tracing and annotating presynaptic segments with at least 3 synapses. We selected consistent partners based on their presence in at least 5% of all analyzed columns. The neurotransmitter identity for these presynaptic partner types was verified through electron microscopy predictions[49], complemented by an RNA seq dataset[41].

## Area and Column Span Calculation
For each Tm9, we identified each connected presynaptic neuron (e.g., Dm12) and calculated the area and column span. For each connecting pair, we initially determined the volume occupied by the given presynaptic neuron perpendicular to the Tm9 transmedullar axis of projection. To measure the area ($\mu m^2$) covered by a presynaptic neuron, we extrapolated the volume into a single plane perpendicular to the Tm9 projection axis and measured it using a convex hull measurement. The Dm12 column span was calculated as the distance between the furthest two points of the volume normalized by the distance of one single column span (averaging 11.2 μm in diameter).

To obtain both the Tm9 projection axis and the plane perpendicular to it for the mentioned procedure, we used the XYZ coordinates for all presynaptic sites of a given Tm9. Since Tm9 is a unicolumnar and transmedullar neuron perpendicular to the medulla surface, the eigenvector of the first PCA component aligns with that axis, and the eigenvectors for the second and third PCA components form the perpendicular plane. All presynaptic neuron XYZ sites were projected to that 2D plane to calculate the covered area for each presynaptic partner.

## Statistics and reproducibility

Sample-sizes for imaging experiments were chosen based on typical sample sizes in the field. All sample sizes are provided in the figure legends. The study was not blinded but all genotypes were mixed within imaging sessions.

Micrographs provided in Figs. 1c and 6b are representative for the corresponding genotypes and reproduced in all corresponding imaging experiments performed.

For statistical analysis, quantified variables were first calculated for each ROI and then averaged within and across flies. Statistical tests are reported in each figure. To determine significance in multiple comparisons, we corrected the α using the Bonferroni method. Appropriate statistical tests were chosen after checking for normality using the Shapiro-Wilk test and homogeneity of variances using the Levene's test.

## Reporting summary

Further information on research design is available in the Nature Portfolio Reporting Summary linked to this article.

## Data availability

Source data of this study can be found on Zenodo: https://doi.org/10.5281/zenodo.13327244[90]. Additionally, source data is also provided in spreadsheet format with this paper. Source data are provided with this paper.

## Code availability

The code for data analysis and model simulations can be found in https://github.com/silieslab/Gur-etal-2024.

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

## Acknowledgements

We thank Jonas Chojetzki and Simone Renner for excellent technical assistance, and are grateful to Carlotta Martelli, Christopher Schnaitmann, and members of the Silies lab for critical comments on the manuscript. We thank Aljoscha Nern for providing the Dm12 split-Gal4 line prior to publication. This work has received funding from the European Research Council (ERC) under the European Union's Horizon 2020 research and innovation program (grant agreement No 716512 and No 101045003), and from the German Research Foundation (DFG) through the collaborative research center CRC 1080 (project C06), and through Research Unit FOR 5289 (project P5) to MA.

## Author contributions

Conceptualization: B.G., L.R., M.S.; Methodology: B.G., L.R.; Software: B.G., L.R., S.M.-O., and G.R.-T.; Investigation: B.G., L.R., J.C., F.T., S.M.-O., and G.R.-T. Writing—Original Draft: B.G., L.R., and M.S.; Writing—Reviewing & Editing: B.G., L.R., J.C., F.T., S.M.-O., G.R.-T., and M.S.; Visualization: B.G., L.R.; Supervision: M.S.; Funding Acquisition: M.S.

## Funding

## Competing interests

The authors declare no competing interests.
