## [Peer Review File · Nature Communications]

REVIEWER COMMENTS

Reviewer #1 (Remarks to the Author):

The coding of luminance and contrast is a classic topic in sensory physiology, so classic that some might regard it as moribund. This paper shows how the topic can be revived, and indeed placed at the forefront of a new sensory neuroscience enabled by the dazzling array of techniques available in *Drosophila* today. The paper is a tour de force, combining calcium imaging, glutamate imaging, knockout of neurotransmitter receptor with cell type specificity, connectomics, and computational modeling to study the mechanisms of invariance to luminance in the fly visual system. It was a great read, and could be published as is. My comments and questions are mostly about clarification. Whether the authors address them should be considered optional.

Figs. 1 and 2 show that visual responses of some cells are invariant to luminance, while responses of other cells are not. This is a simple and striking result. Have such experiments never been done before for T4/T5 and the downstream LPTCs? The latter have been studied for decades.

The claim that Dm12 is glutamatergic is plausible. However, the authors should be cautious about citing evidence from FlyWire. GABA and glutamate are scored with similar probability by FlyWire/Eckstein et al. though not with high confidence.

Why do the authors study the F1 response only? In Fig. 1d, it looks like the F0 response of L3 is more invariant to luminance than the F1 response. Is it that we can't trust the F0 response, because this is calcium imaging?

How specific is the GluSnFR signal? Is it clear that it comes only from glutamate released at synapses onto Tm1 and Tm9?

Abbreviating "luminance gain control" as "luminance gain" seems confusing to me.

In the experiments, luminance is varied over a factor of 10. How was this range chosen? The natural range extends over many orders of magnitude.

Many bar graphs summarize experiments with luminance sensitivity, which I think is defined as the slope of F1 response versus luminance. Does this definition make sense, given that F1 response is roughly linear with log luminance?

It's confusing that the curve for 100% contrast in Figure 2i is different from the Tm1 curve in Figure 2e. Presumably this has something to do with the normalization? The sets of flies/cells are nonoverlapping or overlapping?

In Figure 2, it's fantastic to have recordings from both axon and dendrites.

I was impressed by the demonstration (Fig. 5) that glutamatergic input has a wider spatial range than the receptive field. Very nice result.

In Figures 5h and 5i, the FWHM of the spatial filter fit to the calcium signal is wider than measured by Arenz et al. (2017). The filters are measured using the same technique of white noise analysis. Do the authors know why the results are different?

I was confused by the difference in vertical scale between Figure 6d vs. Figs. 2k and 2l. I finally realized that the latter are normalized. This fact is buried in the Methods, and could be worth mentioning in the Figure legend. Right now it's only noted by "(norm.)" in the y axis label.

The authors show that L3 activity is pooled by Dm12, which in turn inhibits Tm9. This pathway is hypothesized to be the mechanism of luminance gain control. It is also known that Dm12 synapses even more strongly onto L3 than onto Tm9. (Fig. 3a of bioRxiv preprint by Seung, "Interneuron diversity and normalization specificity in a visual system.") It's interesting that Dm12 is unable to make L3 invariant to luminance, but achieves that with Tm9.

Is it also possible that cellular biophysics could be contributing to invariance?

Reviewer #2 (Remarks to the Author):

How visual systems are able to consistently extract particular visual features, such as motion, from an environment where other visual parameters are continuously changing is a broadly relevant question for both biological and in silico visual circuits. Here, Gur et al locate the first emergence of luminance invariance in the early stages of fly visual processing. They combine thoughtfully designed visual stimuli and Ca²⁺ imaging to identify Tm1 and Tm9 neurons are capable of encoding contrast independent of luminance – either stably encoding across luminance (Tm1) or enhancing contrast in low luminance environments (Tm9). They then identify biologically plausible mechanisms by utilizing modeling, connectomics and genetic manipulations to uncover widefield neurons that locally pool luminance values and feed this information back through inhibition onto the medulla neurons.

Strengths:

1. The paper is clearly written and easy to follow
2. The paper incorporates modeling early on and then uses the results from the modeling experiments to test, in vivo, model predictions
3. The paper well utilizes the genetics and connectomics prowess of Drosophila.
4. The paper identifies neural substrates capable of performing specific visual computations and then takes the next step to identify potential mechanisms.

Major Comments:

1. The imaging data support that gain control first happens within a subset of tramedullary neurons, and these experiments are well designed.
2. The authors claim Tm1 and Tm9 pass luminance controlled information to direction selective cells. This can be speculated from connectivity, but silencing of Tm1 and Tm9 and recording from T5 would be required to actually test this. The authors also need to discuss how luminance invariance is maintained in T5 when it receives inputs that are luminance controlled but also inputs that are not luminance controlled (Tm2 and Tm4)
3. Dm12 is predicted to be the source of inhibition, pooling luminance information across multiple columns, and feeding back on to Tm9. Data in figure 7 support it inhibits Tm9 and that it pools information across multiple columns. But no data are shown to demonstrate that loss (silencing) of Dm12 changes Tm9 responses, preventing enhancement of contrast responses at low luminance levels.
4. The authors should discuss the biological implementation of the nonlinearity that was necessary for the inhibitory feedback in the Tm9 model. This nonlinearity would exist within Dm12? Is this aligned with the actual Dm12 luminance responses, and does this make sense given Dm12 neurons may not be spiking neurons? The nonlinearity doesn't seem to be present at the level of glutamate released onto Tm9, which looks similar to glutamate release on Tm1 (Figure 5k).

Minor Comments:

1. LN 314 – “Stable contrast estimation and is first”

Reviewer #2 (Remarks on code availability):

Code is available - I looked through it but did not try to implement it.

Reviewer #3 (Remarks to the Author):

In visual systems, contrast is calculated as the variation in luminance relative to background, which fluctuates with changing light conditions. However, it remains unclear how animals maintain stable contrast perception under constantly changing luminance conditions. In this manuscript, Gur et al used in vivo two-photon imaging and computational approach to investigate the cellular and circuitry mechanisms governing post-receptor luminance gain control in *Drosophila*. The authors found that the dendrites of the third-order visual neurons, TM1 and Tm9, are the initial site where luminance gain control is implemented. Furthermore, they identified a wide-field neurons DM12 that pooled spatial luminance and provided normalization to Tm neurons through glutamate-gated GluCl α channel-mediated inhibition in Tms.

This study is elegantly conducted with solid results and a compelling conclusion.

To enhance the manuscript, the authors could address the following concerns.

1) While Tm1 exhibits stable contrast responses within the relative narrow illuminance range (~10 fold) examined by the authors, it would be interesting to examine Tm1's responses to gradual reduction in illuminance. Do the responses exhibit a step function (from no response to a sudden full response) or a gradual increase in contrast responses? These transition results could provide valuable insights into luminance gain control.

2) How can the increased responses of Tm9 at low luminance intensities be interpreted in light of the GluCl α mechanism?

Reviewer #1 (Remarks to the Author):

The coding of luminance and contrast is a classic topic in sensory physiology, so classic that some might regard it as moribund. This paper shows how the topic can be revived, and indeed placed at the forefront of a new sensory neuroscience enabled by the dazzling array of techniques available in *Drosophila* today. The paper is a tour de force, combining calcium imaging, glutamate imaging, knockout of neurotransmitter receptor with cell type specificity, connectomics, and computational modeling to study the mechanisms of invariance to luminance in the fly visual system. It was a great read, and could be published as is. My comments and questions are mostly about clarification. Whether the authors address them should be considered optional.

Thank you very much for this very positive assessment of our work and your suggestions for improving it.

Figs. 1 and 2 show that visual responses of some cells are invariant to luminance, while responses of other cells are not. This is a simple and striking result. Have such experiments never been done before for T4/T5 and the downstream LPTCs? The latter have been studied for decades.

As pointed out by the reviewer, there is extensive literature on T4/T5 (mostly in *Drosophila*) and on LPTC responses (in *Drosophila*, and a lot more in other flies such as blowflies). For T4/T5, the local motion detectors, contrast adaptation has been studied in these cells and their presynaptic inputs (Drews et al. 2020 - <https://doi.org/10.1016/j.cub.2019.10.035> , Matulis et al. 2020 - <https://doi.org/10.1016/j.cub.2019.11.077>), but experiments assessing luminance-invariant responses have not been done before our study.

Most LPTCs are sensitive to optic flow which should also be encoded independently of environmental luminance. There is no work that assessed luminance responses of *Drosophila* LPTCs. In bigger flies, LPTC responses upon changing the mean luminance of a moving pattern have been measured. In luminance conditions typical for the day (over a certain darkness), blowfly H1 neurons show luminance-invariant response properties when stimulated with a moving pattern, but the time scales of this experiment are not given, so these might be slow or fast gain control mechanisms contributing to this (Eckert 1980 *J Comp Physiol A* - <https://doi.org/10.1007/BF00660179> , Hausen 1984 - https://doi.org/10.1007/978-1-4613-2743-1_15). This is in line with our data, since LPTCs are downstream of the visual circuitry that we found to be responsible for establishing luminance gain control. We now added this information and the citation to the discussion.

The claim that Dm12 is glutamatergic is plausible. However, the authors should be cautious about citing evidence from FlyWire. GABA and glutamate are scored with similar probability by FlyWire/Eckstein et al. though not with high confidence.

Thank you for raising this concern, we have supplemented this claim by additionally referring to Davis et al. 2020 - <https://doi.org/10.1101/385476>, where Dm12 is identified as glutamatergic based on transcriptomic data (it expresses vGluT but not Gad1).

Why do the authors study the F1 response only? In Fig. 1d, it looks like the F0 response of L3 is more invariant to luminance than the F1 response. Is it that we can't trust the F0 response, because this is calcium imaging?

We have focused our analysis on the F1 responses because we study the contrast encoding component of neural responses. The temporal contrast produced by a moving sinusoidal grating happens at the temporal frequency of the grating (in our study at 1Hz) and thus the F1 amplitude of the neural responses reflect contrast encoding. The F0 response of neurons on the other hand, can reflect their luminance encoding properties. Previous studies have shown that L3 neurons encode luminance and respond stronger to low luminances (Ketkar et al. 2020 - <https://doi.org/10.1016/j.cub.2019.12.038>, 2022 - <https://doi.org/10.7554/eLife.74937>). We have analyzed the F0 responses of L3 neurons imaged in our study (Figure R1) and found a similar inverted relationship with luminance as shown before.

Figure R1: L3 F0 responses encode luminance. **Left)** Another example L3 neuron (different from manuscript Fig. 1d) emphasizing that the F0 component follows grating luminance with an inverse relationship. **Right)** F0 component (mean response of the grating epoch) of L3 responses for each luminance level. Error bars represent \pm SEM. Means are calculated across flies. $n=8(150)$ as flies(cells).

How specific is the GluSnFR signal? Is it clear that it comes only from glutamate released at synapses onto Tm1 and Tm9?

The glutamate signal is measured using cell-type specific expression of the glutamate sensor iGluSnFR either in Tm1 or Tm9 neurons. iGluSnFR is thus present at the membrane and we have specifically imaged the iGluSnFR signal at the dendrites of Tm1 and Tm9 neurons located in medulla layers M2-M3, arguing that our data represents glutamate coming specifically onto the dendrites of Tm1 and Tm9 neurons. We now state this (cell-type specific expression and dendritic recordings) explicitly in the text.

Abbreviating "luminance gain control" as "luminance gain" seems confusing to me.

We changed all instances to "luminance gain control".

In the experiments, luminance is varied over a factor of 10. How was this range chosen? The natural range extends over many orders of magnitude.

Our experiments were conducted in illumination conditions where fruit flies have peak activity during the day (Rieger et al. 2007 <https://doi.org/10.1177/0748730407306198> ; Spitschan et al. 2016 <https://doi.org/10.1038/srep26756>) and exhibit luminance-invariant optomotor behavior (Ketkar et al. 2020, <https://doi.org/10.1016/j.cub.2019.12.038>). We are technically limited by our stimulus projector for the two photon imaging paradigm and thus cannot easily extend the range within single experiments. However, previous experiments from the lab have shown that fly behavior is luminance-invariant over 3-4 orders of magnitude and rapid post-receptor luminance gain control is important throughout this wide range of luminance (Ketkar et al. 2020 <https://doi.org/10.1016/j.cub.2019.12.038>). This suggests that our findings will likely generalize over different ranges of illumination conditions. We have now addressed this point in our discussion starting.

Many bar graphs summarize experiments with luminance sensitivity, which I think is defined as the slope of F1 response versus luminance. Does this definition make sense, given that F1 response is roughly linear with log luminance?

That is a valid point, and we agree that it is more suitable to extract slopes from F1 vs log-luminance since the linear-relationship is present with log-luminance. We have re-analyzed our data and now show the F1 vs log-luminance slope in our plots when we show luminance sensitivity of neurons. We have changed the figure legends and methods accordingly. After re-analyzing data and performing statistical tests, our conclusions remain the same. Thank you for the feedback.

It's confusing that the curve for 100% contrast in Figure 2i is different from the Tm1 curve in Figure 2e. Presumably this has something to do with the normalization? The sets of flies/cells are nonoverlapping or overlapping?

These data are coming from different experiments with non overlapping flies. In both cases the data are normalized but in the experiments where we presented gratings with 5 different contrast and 5 different luminance (Fig. 2h-l) the data is a bit more noisy (these are much longer recordings than the ones for Fig. 2a-f). Thus the peak response of individual neurons, which we used for normalizing their data, are not always at the same luminance, especially when the neurons exhibit luminance gain control (like Tm1 and Tm9). However, the major point relevant for our conclusions is that the slope of the curves are similar indicating the presence of luminance gain control in both conditions.

In Figure 2, it's fantastic to have recordings from both axon and dendrites.

Thank you very much for the positive feedback.

I was impressed by the demonstration (Fig. 5) that glutamatergic input has a wider spatial range than the receptive field. Very nice result.

Thank you.

In Figures 5h and 5i, the FWHM of the spatial filter fit to the calcium signal is wider than measured by Arenz et al. (2017). The filters are measured using the same technique of white noise analysis. Do the authors know why the results are different?

The FWHM of the calcium signal measured at the axon terminals of Tm1 and Tm9 is around 10 degrees (Fig. 5h and i). Arenz et al. 2017 (<https://doi.org/10.1016/j.cub.2017.01.051>) quantified the FWHM (center) for Tm1: ~8 degrees and Tm9: ~7 degrees. We believe that this ~2 degrees of difference can come from differences in the white noise stimulation parameters and how the receptive fields are extracted. First of all, we used stripes with a width of 5 degrees whereas Arenz et al. used stripes with a width of 2.8 degrees. Secondly, the brightness values presented in our paradigm differs from Arenz et al. which could affect the width of visual receptive fields (Stöckl et al. 2020 <https://doi.org/10.1126/sciadv.aaz8645>). Furthermore, for data analysis, Arenz et al. fitted a difference of gaussians to model the center-surround of receptive fields and we fitted a single gaussian since we haven't observed a strong surround in white noise extracted RFs. All of these factors could contribute to the differences, which are however rather subtle.

Importantly, we focus on making comparisons between calcium and glutamate RFs and these experiments used identical white noise stimulation and analysis protocols.

I was confused by the difference in vertical scale between Figure 6d vs. Figs. 2k and 2l. I finally realized that the latter are normalized. This fact is buried in the Methods, and could be worth mentioning in the Figure legend. Right now it's only noted by "(norm.)" in the y axis label.

We have clarified this by adding the information where data are normalized to the figure legends.

The authors show that L3 activity is pooled by Dm12, which in turn inhibits Tm9. This pathway is hypothesized to be the mechanism of luminance gain control. It is also known that Dm12 synapses even more strongly onto L3 than onto Tm9. (Fig. 3a of bioRxiv preprint by Seung, "Interneuron diversity and normalization specificity in a visual system.") It's interesting that Dm12 is unable to make L3 invariant to luminance, but achieves that with Tm9.

This is definitely a very interesting point. It might entirely be true that Dm12 synapsing onto L3 is also biologically important, but that it achieves a different kind of normalization which leads to the non-linear luminance encoding of L3 rather than achieving luminance invariance at the contrast

responses. Photoreceptor luminance encoding is linear and normalization by L3 neurons could be mediated by L3-Dm12 feedback connections. This form of normalization could be useful for separating feature encoding such as the distinct encoding of luminance and contrast for the lamina neurons. This idea is also discussed in a recent preprint by Seung (bioRxiv, 2024, <https://doi.org/10.1101/2024.04.03.587837>).

A key player here that differentiates the Dm12 mediated normalizations onto L3 and Tm9 could be GluClalpha expression and/or localization since it is required for generating luminance gain control for Tm9 neurons (Fig. 6). L3 gives input also to other Tm neurons and achieving luminance gain control early on in L3 can affect other processes. For example contrast and luminance gain control need to be controlled independently due to the statistics of natural scenes (Mante et al. 2005 - <https://doi.org/10.1016/j.neuron.2008.03.011>), and adding luminance gain control to L3 might disrupt contrast gain present in other Tm neurons (also see paragraph in the discussion on why luminance gain control might best be implemented in the medulla rather than in photoreceptors or LMCs).

Is it also possible that cellular biophysics could be contributing to invariance?

Yes, cellular biophysics will likely play a major role in luminance gain control. In Fig. 5 a-f we've built a biophysical model of Tm1 and Tm9 neurons which implemented shunting inhibition leading to normalization of contrast responses. We first modeled LMC contrast responses as the major inputs of Tm neurons, and then, we included shunting inhibition via an inhibitory wide-field neuron that pools the signal from neighboring LMC neurons. For Tm9, we then showed that this biophysical mechanism can be implemented by GluClalpha (Fig. 6). In line with our conclusions, GluClalpha was previously indicated for shunting in fly motion circuits (Groschner et al. 2022, <https://doi.org/10.1038/s41586-022-04428-3>).

Reviewer #2 (Remarks to the Author):

How visual systems are able to consistently extract particular visual features, such as motion, from an environment where other visual parameters are continuously changing is a broadly relevant question for both biological and in silico visual circuits. Here, Gur et al locate the first emergence of luminance invariance in the early stages of fly visual processing. They combine thoughtfully designed visual stimuli and Ca²⁺ imaging to identify Tm1 and Tm9 neurons are capable of encoding contrast independent of luminance – either stably encoding across luminance (Tm1) or enhancing contrast in low luminance environments (Tm9). They then identify biologically plausible mechanisms by utilizing modeling, connectomics and genetic manipulations to uncover widefield neurons that locally pool luminance values and feed this information back through inhibition onto the medulla neurons.

Strengths:

1. The paper is clearly written and easy to follow
2. The paper incorporates modeling early on and then uses the results from the modeling experiments to test, in vivo, model predictions

3. The paper well utilizes the genetics and connectomics prowess of *Drosophila*.
4. The paper identifies neural substrates capable of performing specific visual computations and then takes the next step to identify potential mechanisms.

Thank you very much for your assessment of our work, highlighting its strengths (above) and providing suggestions for improvement (below).

Major Comments:

1. The imaging data support that gain control first happens within a subset of transmedullary neurons, and these experiments are well designed.

Thank you for the positive feedback.

2. The authors claim Tm1 and Tm9 pass luminance controlled information to direction selective cells. This can be speculated from connectivity, but silencing of Tm1 and Tm9 and recording from T5 would be required to actually test this. The authors also need to discuss how luminance invariance is maintained in T5 when it receives inputs that are luminance controlled but also inputs that are not luminance controlled (Tm2 and Tm4)

The reviewer is correct in that we have not supported this claim by experiments that establish causality. However, Tm9 is one of the major inputs to T5 (https://doi.org/10.7554/eLife.40025) and silencing Tm9 leads to a strong deficit in T5 neuron responses and in T5 direction selectivity (Fisher et al. 2015, Fig. 5c-h, https://doi.org/10.1016/j.cub.2015.11.018). In line with this, blocking either Tm1 or Tm9 or both neurons, also leads to major deficits in downstream LPTC OFF motion responses (Serbe et al. 2016, http://dx.doi.org/10.1016/j.neuron.2016.01.006). This deficit in T5 neurons/LPTCs in the silenced conditions might challenge the study of luminance gain control since core motion computation is affected. We have done a comprehensive characterization of luminance gain control for all the major inputs of T5 neurons that receive feedforward visual input from photoreceptors (Shinomiya 2019, https://doi.org/10.7554/eLife.40025) which places Tm1 and Tm9 as the main candidates for providing luminance gain controlled input to T5 neurons.

We toned down our statement in the discussion and now say: *The postsynaptic direction-selective T5 neurons receive their major inputs from Tm1, Tm2, Tm4 and Tm9 neurons^{35,36} and maintain luminance-invariant signal. Thus, the luminance-invariant signal in T5 likely originates in Tm1 and Tm9 neurons...*

Second, it is a very interesting question how parallel inputs coming from neurons with luminance gain control (Tm1 and Tm9) and neurons with no luminance gain control but contrast gain control (Tm2 and Tm4) are combined to ensure stable motion processing. This point would require more detailed investigations on T5 neurons. A similar problem exists in the vertebrate LGN where contrast and luminance gain control mechanisms are both present. There, these gain control mechanisms are controlled independently (Mante et al. 2005) and a fast adaptation model composed of two independent resistor-capacitor (RC) stages, one for luminance gain control and another for contrast gain control, can account for LGN responses for both artificial and natural stimuli (Mante et al. 2008, https://doi.org/10.1016/j.neuron.2008.03.011). These RC stages shape the linear receptive field of the LGN neurons to ensure stable responses. While the exact

biophysical and circuit implementation might differ in the fly, it is interesting to know that some solutions have in principle been discussed. We added this to the discussion.

3. Dm12 is predicted to be the source of inhibition, pooling luminance information across multiple columns, and feeding back on to Tm9. Data in figure 7 support it inhibits Tm9 and that it pools information across multiple columns. But no data are shown to demonstrate that loss (silencing) of Dm12 changes Tm9 responses, preventing enhancement of contrast responses at low luminance levels.

In the manuscript, based on EM connectomics, calcium imaging and optogenetic activation we've suggested Dm12 as a candidate involved in achieving luminance gain control in Tm9. Yet, Tm9 neurons exhibit variability in their input distributions (Cornean et al. 2024 - <https://doi.org/10.1038/s41467-024-45971-z>). Even though Dm12 is the 4th most common input of Tm9 neurons, Tm9 neurons receive inputs from Dm12 neurons in only ~45% of all columns (Cornean et al. 2024) suggesting that luminance gain circuit implementation might be distributed across different Tm9 inputs. In line with this idea, different neurons that span multiple columns (e.g., Tm16, C2, C3, etc. Cornean et al. 2024) and also including other Dm neurons (Figure R2) provide heterogenous input to Tm9, and we did not necessarily expect a phenotype from silencing Dm12 alone.

Figure R2: Other Dm neuron inputs presynaptic to Tm9 that could support luminance-invariant visual processing.

To satisfy the reviewer's and our own curiosity, we did the Dm12 silencing experiments. As expected from the variable input structure of Tm9, we did not observe a phenotype in Tm9 luminance gain control when we silenced Dm12 neurons (Fig. R3). Together, variability of Tm9 input connectivity (Cornean et al. 2024) and functional luminance gain control properties (Figure R3), but also other variable receptive field properties that we had described earlier (Ramos-Traslosheros et al. 2021 - <https://doi.org/10.1038/s41467-021-24986-w>) along with our silencing experiments (Figure R4) suggests that Tm9 neurons form a functionally heterogeneous cell population where luminance gain control is implemented via distributed input pathways to achieve

robust visual processing (see also Cornean et al. 2024). We are very excited to continue these heterogeneous properties further but think that this will be another PhD or postdoctoral project.

To make this point about variability more explicit in the manuscript, we now say at the beginning of the corresponding results paragraph: “Tm9 has been shown to have variable presynaptic connectivity in different columns of the eye (Cornean et al. 2024), so this pooling task is likely distributed across different cell types. To identify neurons that can contribute to the task we used the Flywire connectome...”

We also added a section to the discussion “However, Dm12 only provides input to Tm9 in 45% of columns, arguing that other neurons will likely contribute to luminance gain control in other columns. This is interesting, because other wide field neurons could act across other spatial scales. Such a heterogeneity could ensure that different spatial scales present in natural scenes are accommodated across the fly eye.”

Fig. R3: Tm9 luminance gain control depends on distributed circuitry. a) Silencing the Dm12, the major wide-field input to Tm9 neurons. b) Silencing Dm12 do not affect luminance gain control in Tm9 neurons.

4. The authors should discuss the biological implementation of the nonlinearity that was necessary for the inhibitory feedback in the Tm9 model. This nonlinearity would exist within Dm12? Is this aligned with the actual Dm12 luminance responses, and does this make sense given Dm12 neurons may not be spiking neurons? The nonlinearity doesn't seem to be present at the level of glutamate released onto Tm9, which looks similar to glutamate release on Tm1 (Figure 5k).

As the reviewer alludes to, the glutamatergic input onto Tm9 is not luminance invariant yet. We therefore agree that this nonlinearity does not exist within Dm12. The nonlinearity present in Tm9 might instead arise from the shunting inhibition provided by GluClalpha. In line with this, a previous study showed that GluClalpha is involved in the non-linear amplification step of motion computation (Groschner et al. 2022 <https://doi.org/10.1038/s41586-022-04428-3>).

We added the following section to our discussion (Line 431): “The source of this enhancement in low luminance for Tm9 neurons could originate from a non-linearity provided by GluCl α which was shown to non-linearly enhance motion responses in downstream neurons⁴³ “

Minor Comments:

1. LN 314 – “Stable contrast estimation and is first”

Thank you for pointing this out, we fixed it.

Reviewer #2 (Remarks on code availability):

Code is available - I looked through it but did not try to implement it.

Reviewer #3 (Remarks to the author):

In visual systems, contrast is calculated as the variation in luminance relative to background, which fluctuates with changing light conditions. However, it remains unclear how animals maintain stable contrast perception under constantly changing luminance conditions. In this manuscript, Gur et al used in vivo two-photon imaging and computational approach to investigate the cellular and circuitry mechanisms governing post-receptor luminance gain control in *Drosophila*. The authors found that the dendrites of the third-order visual neurons, TM1 and Tm9, are the initial site where luminance gain control is implemented. Furthermore, they identified a wide-field neurons DM12 that pooled spatial luminance and provided normalization to Tm neurons through glutamate-gated GluCl α channel-mediated inhibition in Tms.

This study is elegantly conducted with solid results and a compelling conclusion.

Thank you very much for the positive assessment of our work and your suggestions to improve our manuscript.

To enhance the manuscript, the authors could address the following concerns.

1) While Tm1 exhibits stable contrast responses within the relative narrow illuminance range (~10 fold) examined by the authors, it would be interesting to examine Tm1’s responses to gradual reduction in illuminance. Do the responses exhibit a step function (from no response to a sudden full response) or a gradual increase in contrast responses? These transition results could provide valuable insights into luminance gain control.

We agree that we have explored a relatively narrow luminance range, as compared to the wider range that flies will encounter throughout the day. However, we have previously shown that fly behavior is luminance-invariant over 3-4 orders of magnitude, and behavioral responses gradually decrease in very dark luminances, when the fly likely reaches the detection limit of vision. Furthermore, rapid post-receptor luminance gain control is important throughout this wide

range of luminance, and requires the L3 input for all of these conditions (Ketkar et al. 2020 - <https://doi.org/10.1016/j.cub.2019.12.038>). In line with this, L3 encodes luminance information across this wide range of inputs (Ketkar et al. 2023 - <https://doi.org/10.1016/j.cub.2023.05.024>). This suggests that the underlying circuitry, linking L3 to optomotor behavior (Tm neurons included), will likely also generalize over different ranges of illumination conditions. We have added this argument to the discussion starting from line 339.

While we have not analyzed Tm1 responses under many different adaptation states, we have recorded the response of Tm1, and other Tm neurons, to gradual steps in luminance and compared this to the L3 response. The luminance dependence is very similar between e.g., L3 and Tm1, further arguing that this motif will be utilized across conditions (Fig. R4). We are happy to include this figure as a Supplementary Figure.

Figure R4: Tm1 and Tm9 responses have a luminance sensitive plateau. Example calcium traces of single L3 (green), Tm1 (magenta), Tm2 (brown), Tm4 (orange), Tm9 (dark green) axon terminals to a stimulus comprising 7 s full-field flashes varying randomly between eleven different luminances. To the right of each panel, the plateau responses (mean of the last 1 second of the response) are quantified. Sample sizes are given on the plots as flies(cells). All plots show mean±SEM.

2) How can the increased responses of Tm9 at low luminance intensities be interpreted in light of the GluCl α mechanism?

The non-linearity proposed in the normalization model of Tm9 (Fig. 5f) is the important factor for modeling the increased response of Tm9 at low luminance. We argue that this non-linearity arises from GluCl α properties, especially the shunting inhibition provided by GluCl α . In line with this, GluCl α was previously shown to provide a non-linear amplification step in motion circuits (Groschner et al. 2022, <https://doi.org/10.1038/s41586-022-04428-3>). We discuss this now in line 431.

REVIEWERS' COMMENTS

Reviewer #1 (Remarks to the Author):

The authors took great pains to address referees' comments. The revised manuscript is ready for publication.

Reviewer #2 (Remarks to the Author):

The authors have sufficiently addressed my concerns and I believe this manuscript is of high impact and an ideal candidate for Nature Communications.

Reviewer #2 (Remarks on code availability):

code is available

Reviewer #3 (Remarks to the Author):

The authors have adequately addressed the review comments. I recommend publishing it as it is.

Response to reviewer comments for the second round of revisions

We are glad to hear that we have sufficiently addressed the reviewers' points. We thank all the three reviewers for their constructive criticism that improved our manuscript significantly during the review process.

Reviewer #1 (Remarks to the Author):

The authors took great pains to address referees' comments. The revised manuscript is ready for publication.

Reviewer #2 (Remarks to the Author):

The authors have sufficiently addressed my concerns and I believe this manuscript is of high impact and an ideal candidate for Nature Communications.

Reviewer #2 (Remarks on code availability):
code is available

Reviewer #3 (Remarks to the Author):

The authors have adequately addressed the review comments. I recommend publishing it as it is.